# A human mitofusin 2 mutation can cause mitophagic cardiomyopathy

Antonietta Franco[1], Jiajia Li[1], Daniel P Kelly[2], Ray E Hershberger[3], Ali J Marian[4], Renate M Lewis[5], Moshi Song[1], Xiawei Dang[1], Alina D Schmidt[6], Mary E Mathyer[6], John R Edwards[1], Cristina de Guzman Strong[6], Gerald W Dorn[1]*

[1]Department of Internal Medicine, Pharmacogenomics, Washington University School of Medicine, St Louis, United States; [2]Department of Medicine, Cardiovascular Institute, Perelman School of Medicine, University of Pennsylvania, Philadelphia, United States; [3]Department of Internal Medicine, Divisions of Human Genetics and Cardiovascular Medicine, Ohio State University, Columbus, United States; [4]Center for Cardiovascular Genetic Research, University of Texas Health Science Center at Houston, Houston, United States; [5]Department of Neurology, Washington University School of Medicine, St. Louis, United States; [6]Department of Internal Medicine (Dermatology), Washington University School of Medicine, St. Louis, United States

**Abstract** Cardiac muscle has the highest mitochondrial density of any human tissue, but mitochondrial dysfunction is not a recognized cause of isolated cardiomyopathy. Here, we determined that the rare mitofusin (MFN) 2 R400Q mutation is 15–20× over-represented in clinical cardiomyopathy, whereas this specific mutation is not reported as a cause of MFN2 mutant-induced peripheral neuropathy, Charcot–Marie–Tooth disease type 2A (CMT2A). Accordingly, we interrogated the enzymatic, biophysical, and functional characteristics of MFN2 Q400 versus wild-type and CMT2A-causing MFN2 mutants. All MFN2 mutants had impaired mitochondrial fusion, the canonical MFN2 function. Compared to MFN2 T105M that lacked catalytic GTPase activity and exhibited normal activation-induced changes in conformation, MFN2 R400Q and M376A had normal GTPase activity with impaired conformational shifting. MFN2 R400Q did not suppress mitochondrial motility, provoke mitochondrial depolarization, or dominantly suppress mitochondrial respiration like MFN2 T105M. By contrast to MFN2 T105M and M376A, MFN2 R400Q was uniquely defective in recruiting Parkin to mitochondria. CRISPR editing of the R400Q mutation into the mouse *Mfn2* gene induced perinatal cardiomyopathy with no other organ involvement; knock-in of Mfn2 T105M or M376V did not affect the heart. RNA sequencing and metabolomics of cardiomyopathic Mfn2 Q/Q400 hearts revealed signature abnormalities recapitulating experimental mitophagic cardiomyopathy. Indeed, cultured cardiomyoblasts and in vivo cardiomyocytes expressing MFN2 Q400 had mitophagy defects with increased sensitivity to doxorubicin. MFN2 R400Q is the first known natural mitophagy-defective MFN2 mutant. Its unique profile of dysfunction evokes mitophagic cardiomyopathy, suggesting a mechanism for enrichment in clinical cardiomyopathy.

*For correspondence:
gdorn@wustl.edu

**Competing interest:** The authors declare that no competing interests exist.

## Editor's evaluation

The study presents a valuable finding on the link between mitofusin function (MFN2) and Parkin-mediated mitophagy, and that the combination of mitophagy and mitochondrial fusion defects is the basis of cardiomyopathy. The data based on the in vivo models are compelling and provide clinically relevant associations. This finding is important for understanding mitochondrial defects as a basis of heart pathologies.

**eLife digest** Mitochondria are organelles with an essential role in providing energy to the cells of the body. If damaged, they are repaired by fusing and exchanging contents with sister mitochondria in a process that requires mitofusin proteins.

While mutations in the gene for mitofusin 2 have been linked to nerve damage, they do not appear to affect the heart – despite high concentrations of mitochondria in heart muscle cells. However, previous research showed that experimentally disrupting the programmed removal of mitochondria, a process also regulated by mitofusin 2, can cause heart muscle disease known as cardiomyopathy. This suggests that mutations affecting different mitofusin 2 roles might harm individual cell types in different ways.

To investigate, Franco et al. carried out a genetic screen of people with cardiomyopathy, identifying a rare mitofusin 2 mutation, called R400Q, that was more common in this group. Experiments showed that R400Q caused cardiomyopathy in mice and affected mitochondrial repair and replacement, but not movement. By contrast, a mutation linked to Charcot-Marie-Tooth disease type 2A – which causes nerve damage – affected mitochondrial movement but not clearance, leading to nerve cell damage but not cardiomyopathy. This led Franco et al. to suggest that mitochondrial movement is central to nerve cell health, whereas mitochondrial repair and replacement plays an important role in cardiac development.

Genetic cardiomyopathies affect around 1 in 500 people, but only half of the gene mutations responsible are known. These results suggest that mutations affecting mitochondrial quality control factors could be involved, highlighting a direction for future studies into modifiers of cardiomyopathy.

## Introduction

Hearts rely upon mitochondria-derived ATP to fuel cardiac development, excitation–contraction–coupling, and myocardial repair after damage or senescence. For this reason, cardiac involvement is common in diseases like MELAS and MERRF caused by genetic defects of the mitochondrial genome (mtDNA) (*Hsu et al., 2016*). However, the vast majority of the ~1000 constituent mitochondrial proteins are not encoded by mtDNA, but instead by nuclear genes. Among nuclear-encoded mitochondrial proteins having central roles in mitochondrial health and homeostasis are mitofusins (MFN) 1 and 2 (*Santel and Fuller, 2001*). Gene ablation and mutant transgene expression studies targeting cardiac myocytes indicate that cardiac MFNs are critical mediators of reparative mitochondrial fusion (*Chen et al., 2011*; *Kasahara et al., 2013*), mitochondrial culling for quality control (*Song et al., 2015*; *Song et al., 2017*), and mitochondrial replacement for metabolic plasticity (*Gong et al., 2015*).

Clinical and experimental data relating MFN functioning to cardiac health appear somewhat discordant. On the one hand, the cited *Mfn* gene knockout studies unambiguously demonstrate that mitofusins are essential for heart development, contraction, and repair. On the other hand, cardiac involvement has not been reported in human conditions caused by natural loss-of-function *MFN* gene mutations. Indeed, dozens of different pathological *MFN2* mutations cause neuronal rather than cardiac pathology, manifested as Charcot–Marie–Tooth disease type 2A (CMT2A) (*Züchner et al., 2004*) or optic degeneration (*Züchner et al., 2006*). We have proposed that mitochondrial motility defects induced by mutant MFN2 are responsible for clinical neuropathies because neurons have long axonal and dendritic processes that make them critically dependent upon mitochondrial transport (*Dorn, 2019*; *Dorn, 2021*). By contrast, impaired mitophagy seems to underlie developmental and senescent cardiomyopathies provoked by *Mfn* gene ablation or engineered MFN2 mutant expression (*Bhandari et al., 2014*; *Gong et al., 2015*; *Dorn, 2016*).

MFN2 multifunctionality reflects its ability to bind different protein partners as determined by specific phosphorylation events (*Dorn, 2020*; *Li et al., 2022*). Thus, MFN2 oligomerization with MFN1 or MFN2 mediates mitochondrial fusion (*Chen et al., 2010*; *Dorn and Dang, 2022*), MFN2 binding of Parkin mediates mitophagy (*Chen and Dorn, 2013*; *Li et al., 2022*), and MFN2 binding to Miro regulates mitochondrial transport (*Misko et al., 2010*). We reasoned that mutations affecting different MFN2 domains could evoke a phenotypic spectrum consisting of mitochondrial fragmentation from loss of fusion, mitochondrial dysmotility from impaired transport, and/or mitochondrial degeneration from interrupted mitophagy.

Here, as part of an ongoing effort to understand the relationship between *MFN2* mutation site, protein dysfunction, and disease manifestation, we screened cardiomyopathy research cohorts for MFN2 mutations linked to cardiac hypertrophy or heart failure. A largely overlooked, very rare *MFN2* mutation, MFN2 R400Q (*Eschenbacher et al., 2012*), was over-represented in cardiomyopathy among individuals of African descent. We found that fusogenicity of mutant MFN2 Q400 is depressed to the same extent as CMT2A mutants MFN2 T105M and M376 A/V. Uniquely, MFN2 Q400 is defective in recruiting Parkin to mitochondria, thereby impacting an important mitophagic pathway. Knockin (KI) mice engineered to carry Mfn2 Q400 develop a perinatal cardiomyopathy, whereas Mfn2 T105M KI mice develop a CMT2A-like neuropathy and Mfn2 M376V KI mice appear phenotypically normal. To our knowledge, MFN2 R400Q is the first example of a naturally occurring *MFN2* mutation that primarily impacts mitophagy. These findings designate mitophagic dysfunction as a new mechanism for cardiomyopathy, supporting targeted investigations of other genetic mitophagy defects as cardiac risk modifiers. Finally, our data support the concept that different patterns of mitofusin dysfunction underlie organ-specific manifestations of clinical diseases caused by *MFN2* mutations.

## Results

### The rare MFN2 R400Q mutation is over-represented in clinical cardiomyopathy

Rare mutations can be overlooked in standard genotype–phenotype association studies, requiring targeted analyses of exome/whole-genome sequence data (*Auer and Lettre, 2015*; *Bomba et al., 2017*). Here, to define the landscape of potential disease-causing *MFN2* variants (*MacArthur et al., 2012*) we surveyed the gnomAD database (v2.1.1) of 141,456 individuals for rare (minor allele frequency [MAF] of 0.001–0.01) nonsynonymous MFN2 sequence variants. A total of 352 rare nonsynonymous *MFN2* variants are reported (MAF 3.98E-06–6.87 E-03) of which 171 are singletons. Notably, 76 of the rare *MFN2* variants are implicated in CMT2A (*Stuppia et al., 2015*; *Figure 1—source data 1*).

To detect any *MFN2* variants associated with human heart conditions, *MFN2* gene exons were resequenced in DNA samples from 398 subjects with idiopathic/non-ischemic dilated cardiomyopathy (DCM) from the Cincinnati Heart Study (*Liggett et al., 2008*; *Matkovich et al., 2010*; *Cappola et al., 2011*), 286 subjects with hypertrophic cardiomyopathy (HCM) from the Houston Hypertrophic Cardiomyopathy cohort (*Osio et al., 2007*; *Li et al., 2017*), and 861 non-affected controls from the Cincinnati cohort. Additionally, 424 individuals (281 probands) with familial cardiomyopathy (FCM) from the DCM Precision Medicine Study (*Huggins et al., 2022*; *Trachtenberg et al., 2022*) were screened. As summarized in *Table 1*, 11 non-synonymous *MFN2* variants were detected, 6 of which encode putative CMT2A mutations. Also, 8 of the 11 *MFN2* variants were singletons and therefore of minimal interest. MFN2 R400Q, which is not a known CMT2A mutation and is bioinformatically predicted to be damaging, was detected in three unrelated individuals: two with HCM and one with DCM (*Table 1*).

The MFN2 R400Q mutation (*Figure 1A and B*) is reported almost exclusively in African populations in the gnomAD database (*Figure 1—source data 1*). Indeed, all three subjects with the R400Q variant in our cardiomyopathy study populations identified as African American. The overall *MFN2* R400Q MAF of 0.0015 in cardiomyopathy, when compared to MAF of 0.000074 from the gnomAD database, represents ~20-fold over-representation in the combined heart disease populations (p<0.00001). If only the gnomAD African-derived population is used to account for racial differences in R400Q prevalence (10/24,962), over-representation was ~15-fold (p-value = 0.0033). None of the non-affected controls (625 Caucasian and 236 African American subjects) had the MFN2 R400Q mutation. Because a previous description of this *MFN2* mutant described impaired fusogenicity conferring retinal and myocardial abnormalities in transgenic *Drosophila* (*Eschenbacher et al., 2012*), we surmised that, like other loss-of-function MFN2 mutants (*Flannick et al., 2014*; *Steinberg et al., 2015*), MFN2 Q400 could have pathophysiological relevance. We therefore designated it a candidate cardiomyopathy modifier and proceeded with detailed investigations.

### MFN2 Q400 is a dominant suppressor of mitochondrial fusion

Many CMT2A MFN2 mutations, including prototypical MFN2 T105M, are located within the GTPase domain (*Figure 1A*, green region) and lack catalytic GTPase activity (*Franco et al., 2022*). By contrast,

**Table 1.** Results of genetic screening for MFN2 mutations in adult cardiomyopathy.

HCM, hypertrophic cardiomyopathy; DCM, dilated cardiomyopathy; MAF, minor allele frequency. MFN2 protein domains correspond to colored regions in *Figure 1A*.

| MFN2 variant | MFN2 domain | Polyphen 2 | ClinVar | HCM n = 286 | DCM1 n = 398 | DCM2 n = 281 probands (424 cases) | CM MAF n = 965 | gnomAD MAF n = 141,456 | MAF CM vs. gnomAD (p-value) | MAF CM vs. gnomAD (X² stat) |
|---|---|---|---|---|---|---|---|---|---|---|
| S180R | GTPase | Benign | CMT2A | 1 het | | | Singleton | 1.591 E-05 | | |
| V181M | GTPase | Prob dam | CMT2A | | | 1 family; 5 indiv | Singleton | 3.976 E-06 | | |
| S263Y | GTPase | Pos dam | | | 1 het | | Singleton | - | - | - |
| R274Q | GTPase | Benign | CMT2A | | | 1 family; 4 indiv | Singleton | 2.486 E-05 | | |
| G298R | GTPase | Benign | CMT2A | 1 het | | | Singleton | 2.147 E-03 | | |
| M393I | HR1 | Benign | Likely benign | 1 het | | | Singleton | 1.591 E-05 | | |
| R400Q | HR1 | Prob dam | Uncertain significance | 2 hets | 1 het | | 1.55 E-03 | 7.423 E-05 | <0.00001 | 89.595 |
| Q413P | HR1 | Benign | | | | 1 family; 1 indiv | Singleton | 1.193 E-05 | | |
| R468H | inter | Pos dam | CMT2A | 1 het | | 2 families; 5 indiv | 1.55 E-03 | 2.177 E-03 | NS | 0.3431 |
| V705I | HR2 | Benign | Likely benign | 2 hets | | 6 families; 15 indiv | 4.14 E-03 | 6.865 E-03 | NS | 2.0848 |
| A716T | HR2 | Pos dam | CMT2A | | | 1 family; 1 indiv | Singleton | 1.308 E-04 | | |

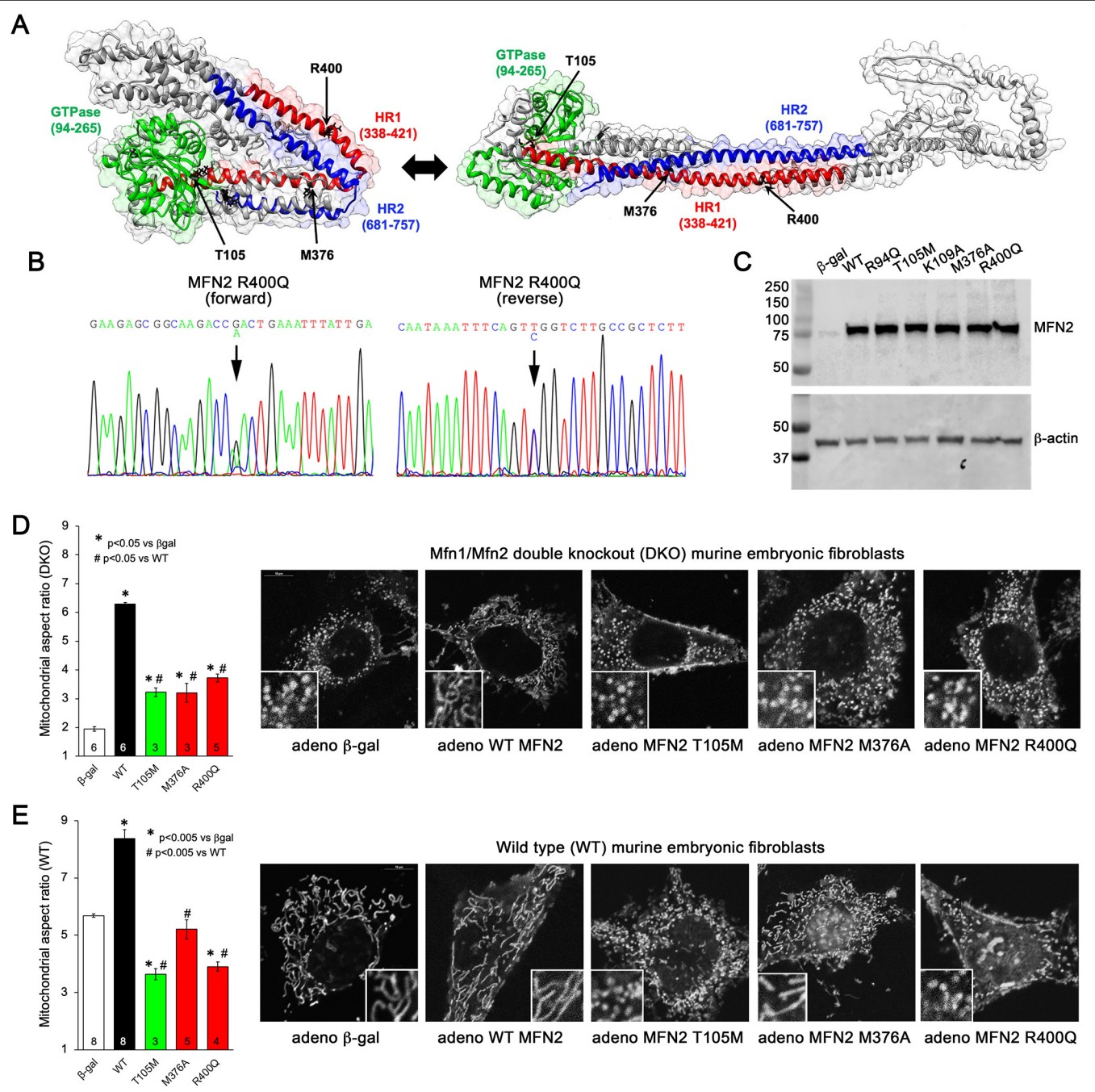

**Figure 1.** Morphological effects of disease-related MFN2 mutants on mitofusin null and normal mitochondria. (**A**) Hypothetical structures of human MFN2 in the basal/closed (left) and active/open (right) conformations. Domains are color-coded: green, GTPase; red, first heptad repeat (HR1); blue, second heptad repeat (HR2). Arrows indicate positions of human disease-linked MFN2 mutants of primary interest. (**B**) Sequencing pherogram from a hypertrophic cardiomyopathy subject showing heterozygous MFN2 R400Q mutation (arrow). (**C**) Immunoblot of wild-type (WT) and mutant MFN2s expressed in Mfn1/Mfn2 double knockout (DKO) murine embryonic fibroblasts (MEFs). In all studies, adenoviral β-gal is negative control; β-actin is loading control. (**D**) Intrinsic fusogenic activity of WT and mutant MFN2 expressed in MFN DKO cells, assayed as increase in mitochondrial aspect ratio compared to β-gal. (**E**) Same as (**D**) except MFNs were expressed in WT MEFs to reveal effects on endogenous MFN1 and MFN2. Unless otherwise stated, in all graphs black is WT MFN2, green is GTPase domain MFN2 mutants, red is HR1 domain MFN2 mutants. Numbers in bars are independent experiments for 6–30 cells per experiment. p-Values used ANOVA and Tukey's test.

The online version of this article includes the following source data and figure supplement(s) for figure 1:

**Source data 1.** gnomAD MFN2 coding variants.

*Figure 1 continued on next page*

*Figure 1 continued*

**Source data 2.** Mitofusin 2 (75 kDa) and β-actin (45 kDa) expression level in MEFs Wt.

**Figure supplement 1.** Fusogenic activities of GTPase-defective, CMT2A-linked MFN2 R94Q, and laboratory-engineered K109A in Mfn1/Mfn2 double knockout (DKO) murine embryonic fibroblasts (MEFs) (left) and WT MEFs (right).

MFN2 R400Q (*Figure 1B*) is located in the hydrophobic core within the so-called first heptad repeat domain (HR1; *Figure 1A*, red region). To better understand the relationship between *MFN2* mutation location, protein dysfunction, and pathophysiological consequence, we compared fusogenic activities of mutant MFN2 Q400 to wild-type (WT) MFN2 and two positionally diverse human CMT2A mutants, MFN2 T105M in the GTPase domain and MFN2 M376V/A in the HR1 domain (*Figure 1A*; *Franco et al., 2020*; *Zhou et al., 2021*). An additional CMT2A GTPase domain mutation, MFN2 R94Q, and laboratory-engineered MFN2 K109A (*Detmer and Chan, 2007*), were included as complementary comparators (see *Figure 1*, *Figure 1—figure supplement 1*). Each human MFN2 was expressed at equal levels in murine mitofusin null (*Mfn1/Mfn2* double knockout [DKO]) cells using adenoviral transduction (*Figure 1C*; *Li et al., 2019*). Importantly, using this system the transfected mitofusin proteins are properly inserted into mitochondria and fully functional (*Franco et al., 2022*).

Mitochondrial aspect ratio (length/width) is commonly used to gauge the relative activity of mitochondrial fusion vs. fission. The normal mitochondrial aspect ratio of ~5–6 in WT murine embryonic fibroblasts (MEFs) is reduced to ~2 in MEFs lacking both mitofusins (Mfn DKO) (compare white bars in *Figure 1D and E*). An increase in aspect ratio in Mfn DKO cells reflects intrinsic fusogenic activity of recombinant MFN2 proteins expressed therein (*Franco et al., 2016*). Consistent with an absolute requirement for GTP hydrolysis during mitochondrial fusion, all three GTPase domain MFN2 mutants exhibited diminished fusogenicity compared to WT MFN2 (*Figure 1D*, *Figure 1—figure supplement 1*; left panel, green bars). Intrinsic fusogenicity of MFN2 M376A and R400Q was similarly impaired (*Figure 1D*, red bars).

Most CMT2A-associated MFN2 mutants do not simply exhibit reduced intrinsic fusogenicity, but dominantly suppress mitochondrial fusion mediated by normal mitofusins 1 and 2. We assayed suppression of mitochondrial fusion by expressing our panel of MFN2 mutants in WT MEFs, which have normal endogenous Mfn1 and Mfn2 and, therefore, a characteristic baseline mitochondrial aspect ratio of 5–6 (*Figure 1E*, white bar). As expected, expression of WT MFN2 caused mitochondrial hyper-fusion, increasing mitochondrial aspect ratio to ~8 (*Figure 1E*, black bar). By comparison, all three GTPase domain MFN2 mutants and MFN2 R400Q suppressed normal mitochondrial fusion (i.e. lowered mitochondrial aspect ratio vs. adeno-βgal transduced) in WT cells. Uniquely among the expressed MFN2 mutants, MFN2 M376A did not alter baseline aspect ratio of WT MEF (*Figure 1E*, *Figure 1—figure supplement 1*, right panel). Thus, MFN2 R400Q is similar to GTPase domain MFN2 mutants in that it both lacks intrinsic fusogenic activity and dominantly suppresses normal mitochondrial fusion, whereas MFN2 M376A is a straightforward functional null protein without dominant inhibitory effects.

## MFN2 HR1 mutants have normal GTPase activity and do not impair mitochondrial respiration

MFN1 and MFN2 are dynamin family GTPases; catalytic GTPase activity is required for mitochondrial fusion (*Chan, 2006*; *Santel and Fuller, 2001*; *Franco et al., 2022*). We asked how suppression of fusion by our panel of MFN2 mutants related to differences in GTPase activity. Adenovirally expressed WT MFN2 accounted for approximately two-thirds of total mitochondrial GTPase activity in transduced mitofusin DKO cells (*Figure 2A*, black bar). As expected, MFN2 T105M, R94Q, and K109A each lacked GTPase activity (*Figure 2A*, *Figure 2—figure supplement 1*, left panel, green bars). Remarkably, GTPase activity of MFN2 R400Q and M376A was the same as WT MFN2, that is, normal (*Figure 2A*, red bars). Thus, diminished GTPase activity cannot be the mechanism for impaired fusogenicity of these two MFN2 HR1 domain mutants.

Mitochondrial respiration and oxidative phosphorylation of ADP to ATP are driven by an electrochemical gradient across the inner mitochondrial membrane, the so-called 'proton pump' (*Madeira, 2018*). Functional electrochemical inner membrane integrity can be assessed using dyes that are concentrated within mitochondria in a voltage-dependent manner; loss of inner membrane electrical

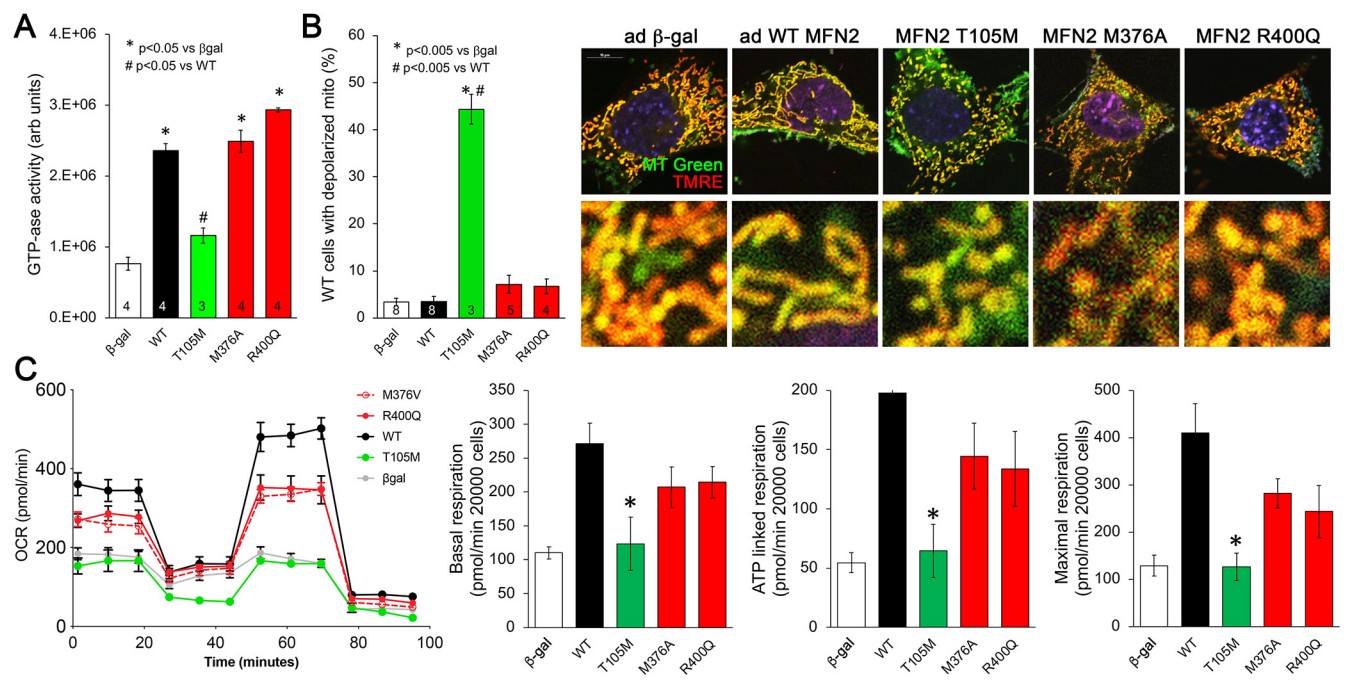

**Figure 2.** Effects of disease-related MFN2 mutants on catalytic and respiratory function of mitofusin null and normal mitochondria. (**A**) GTPase activity of human disease-linked MFN2 mutants expressed in mitofusin double knockout (DKO) murine embryonic fibroblasts (MEFs). (**B**) Loss of inner membrane polarization in WT MEF mitochondria induced by disease-related MFN2 mutants. To the right are representative MitoTracker Green/TMRE co-stained confocal images. (**C**) Seahorse studies of oxygen consumption rate (OCR) in MFN DKO MEFs expressing disease-linked MFN2 mutants. (Left) Average data from three experiments per condition. (Right) Group quantitative results for basal, ATP-linked and maximal respiration (pmol/min/20,000 cells). *p<0.05 vs. WT.p-Values for all studies used ANOVA and Tukey's test.

The online version of this article includes the following figure supplement(s) for figure 2:

**Figure supplement 1.** GTPase activity of, and inner membrane depolarization induced by, GTPase domain CMT2A-linked MFN2 R94Q and laboratory-engineered K109A expressed in Mfn1/Mfn2 double knockout (DKO) murine embryonic fibroblasts (MEFs) (left) and WT MEFs (right), respectively.

polarization abrogates mitochondrial staining. We used tetramethylrhodamine ethyl ester (TMRE) to assess mitochondrial polarization in WT MEFs expressing our panel of MFN2 mutants. Dissipation of the inner membrane electrochemical gradient (mitochondrial depolarization) was provoked by MFN2 T105M, R94Q, and K109A, but not by MFN2 M376A or R400Q (*Figure 2B*, *Figure 2—figure supplement 1*, right panel). Thus, for these MFN2 mutants, the ability to induce mitochondrial depolarization corresponds with impaired catalytic GTPase activity.

The functional consequences of MFN2 mutations on catalytic GTPase activity and electrochemical integrity were further assessed by measuring mitochondrial respiration in DKO MEFs transduced with adeno-MFN2 WT, T105M, M376A, or R400Q. As expected, introducing WT MFN2 into cells lacking mitofusins enhanced all phases of mitochondrial respiration (*Figure 2C*, black). By comparison, MFN2 T105M failed to enhance mitochondrial respiration (*Figure 2C*, green). Strikingly, MFN2 M376A and R400Q were intermediate in their effects, increasing mitochondrial respiration vs. the complete absence of mitofusins (i.e. vs. β-gal), but seemingly not to the same extent as WT MFN2 (*Figure 2C*, red). These data provide further evidence supporting the central relationship between mitochondrial GTPase activity, polarization, and respiration. In the context of our results presented in *Figure 1*, these results implicate a different, unidentified mechanism by which MFN2 HR1 domain mutants impair mitochondrial fusion and suppress mitochondrial respiration.

## MFN2 HR1 domain mutations cause loss of conformational malleability

Mitofusin protein conformation changes upon activation, transitioning from a closed basal state to an open state that favors mitochondrial fusion (*Franco et al., 2016*). The closed conformation is maintained by peptide–peptide interactions involving MFN2 amino acids 367–384 in the HR1 domain (*Rocha et al., 2018*; *Dorn, 2019*). We considered that M376 mutations within the peptide–peptide

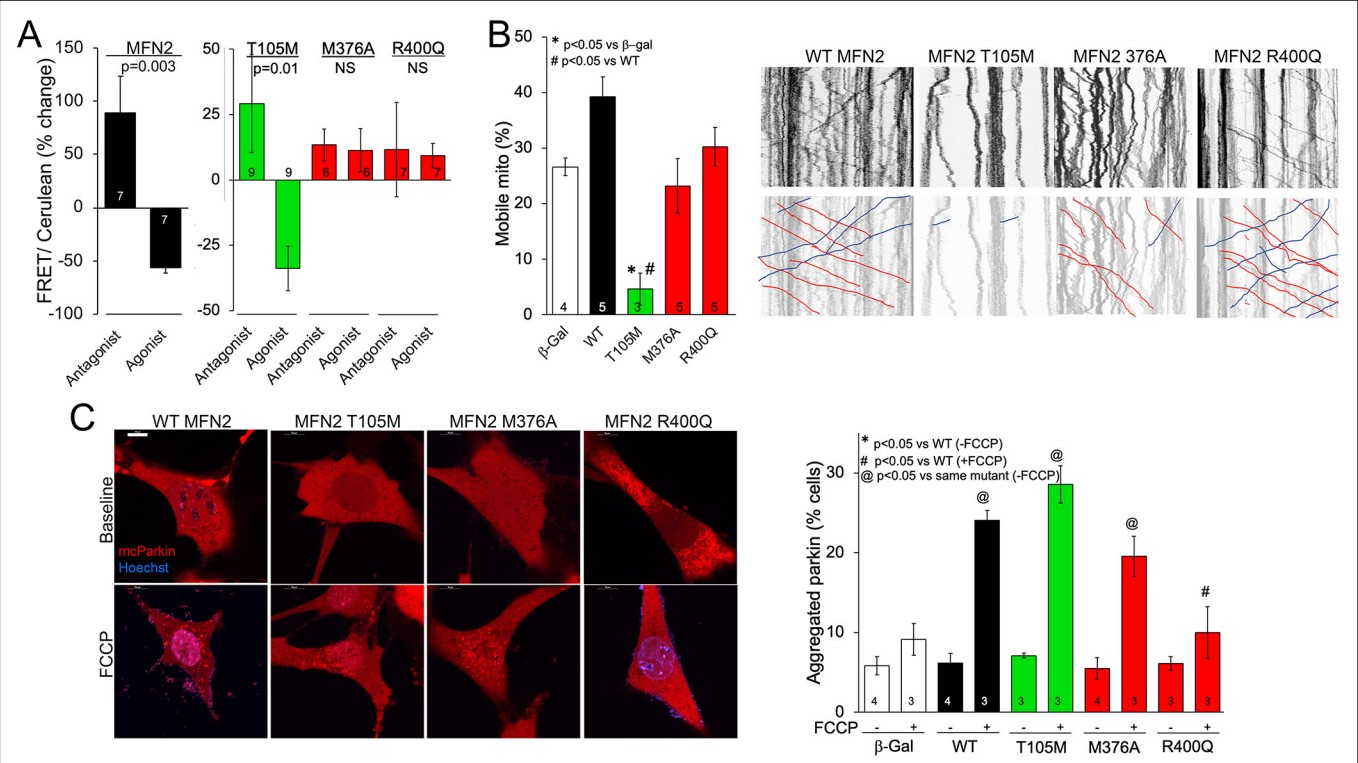

**Figure 3.** MFN2 R400Q impairs conformational shifting and suppresses mitochondrial Parkin recruitment, but not mitochondrial motility in neuronal axons. (**A**) Forster resonance energy transfer (FRET) studies of conformational switching in disease-linked MFN2 mutants expressed in MFN double knockout (DKO) murine embryonic fibroblasts (MEFs). Antagonist peptide normally induces closed conformation (increase in FRET) and agonist peptide open conformation (decrease in FRET). (**B**) Mitochondrial motility in cultured mouse dorsal root ganglion neuronal processes: representative kymographs are to the right (raw data above; motile mitochondrial emphasized below [red, antegrade; blue, retrograde]).Scale bar 10 µm, Data are n~3 5 DRGs for each independent experiment, statistic is one-way ANOVA. (**C**) mcParkin (red) recruitment to mitochondria in MFN DKO MEFs expressing MFN2 mutants before (baseline) and after FCCP (left, representative confocal micrographs; right, group quantitative data). p-Values calculated using ANOVA and Tukey's test.

The online version of this article includes the following figure supplement(s) for figure 3:

**Figure supplement 1.** Effects of engineered synthetic MFN2 mutants on mitochondrial Parkin recruitment and mitochondrial motility in neuronal axons.

interaction domain, and the proximity of the MFN2 R400 mutation site to the interacting domain, might alter the ability of these MFN2 mutant proteins to switch conformation. Consistent with this notion, GTPase-defective CMT2A mutant MFN2 T105M transitioned normally from open to closed state, as measured by antagonist and agonist peptide-induced changes in Forster resonance energy transfer (FRET; *Dang et al., 2020*; *Figure 3A*, green bars). By contrast, the MFN2 M376A and R400Q FRET signals did not change in the presence of MFN antagonist and agonist peptides that normally provoke the closed or open conformation, respectively (*Figure 3A*, red bars; *Franco et al., 2016*). Thus, these MFN2 HR1 domain mutants are defective in their ability to move between open fusion-permissive and closed fusion-limiting conformations, which can explain how mitochondrial fusion is impaired by these mutations despite normal catalytic GTPase activity.

## MFN2 HR1 domain mutations do not impair mitochondrial motility

The appellative function of mitofusins is to mediate mitochondrial fusion, but MFN2 has additional non-canonical tasks including to promote mitochondrial transport (*Misko et al., 2010*; *Rocha et al., 2018*; *Franco et al., 2020*). Indeed, some CMT2A-linked MFN2 mutations dominantly suppress mitochondrial motility, thereby contributing to die-back of long peripheral nerves that require distal mitochondrial delivery for cell renewal and regeneration (*Misko et al., 2010*; *Rocha et al., 2018*; *Franco et al., 2020*). Directed mitochondrial transport can readily be measured in cultured neurons (*Dang et al., 2022*; *Dorn and Dang, 2022*). Accordingly, we expressed our representative MFN2 mutants in

cultured dorsal root ganglion (DRG) neurons derived from wild-type mice. Consistent with previous observations that mitochondrial fusion and motility can be co-regulated by MFN2 (*Franco et al., 2022*), mitochondrial transport through neuronal processes was suppressed by GTPase-defective MFN2 mutants T105M and K109A (*Figure 3B*, *Figure 3—figure supplement 1A*, green bar), and an MFN2 mutant engineered to mimic phosphorylation at sites that suppress mitochondrial fusion, MFN2 EE (*Chen and Dorn, 2013*; *Figure 3—figure supplement 1A*, gray bar MFN2 E111, E442). Mitochondrial motility in DRG neurons was unaffected by HR1 domain mutants M376A and R400Q (*Figure 3B*, red bars) or by an MFN2 mutant engineered to prevent phosphorylation at the fusion-regulatory sites, MFN2 AA (*Chen and Dorn, 2013*; *Figure 3—figure supplement 1A*, A111A442). Thus, combined abnormalities of mitochondrial fusion and motility exhibited by GTPase-defective MFN2 mutants are dissociated in MFN2 HR1 domain mutants.

## MFN2 Q400 is non-functional as a mitochondrial Parkin receptor

MFN2 can initiate mitophagy by switching from a fusion protein into a 'Parkin receptor' that recruits cytosolic Parkin to damaged mitochondria (*Chen and Dorn, 2013*; *Xiong et al., 2019*; *Li et al., 2022*). Mitophagic vs. fusogenic properties are determined by the phosphorylation status of MFN2 T111 and S442 (*Li et al., 2022*). Prior studies have suggested that MFN2 functioning as a Parkin receptor and mitophagy effector is important to heart development and/or contractile function (*Chen and Dorn, 2013*; *Gong et al., 2015*; *Song et al., 2015*). Mitochondrial Parkin recruitment by MFN2 WT, T105M, M376A, and R400Q was compared to that evoked by the MFN2 phosphorylation site mutants MFN2 EE that constitutively bind Parkin and MFN2 AA that is incapable of binding Parkin (*Chen and Dorn, 2013*). Parkin translocation from cytosol to mitochondria was stimulated by uncoupling mitochondrial respiration with carbonyl cyanide p-trifluoromethoxyphenylhydrazone (FCCP) and visualized as the appearance of mitochondrial Parkin aggregates (*Figure 3—figure supplement 1B*). Baseline cytosolic Parkin localization in the absence of FCCP was not affected by WT MFN2 or any of the naturally occurring MFN2 mutants (*Figure 3C*, *Figure 3—figure supplement 1C*, FCCP), whereas MFN2 EE promoted spontaneous Parkin aggregation as expected (*Figure 3—figure supplement 1C*; *Chen and Dorn, 2013*; *Li et al., 2022*). Parkin translocation in response to FCCP was unaffected by MFN2 WT, T105M, K109A, and M376A, but was markedly reduced in cells expressing MFN2 R400Q (*Figure 3C*, *Figure 3—figure supplement 1C*,+FCCP). Indeed, suppression of Parkin recruitment by MFN2 R400Q was similar in magnitude to that of MFN2AA (*Chen and Dorn, 2013*; *Gong et al., 2015*; *Figure 3C*, *Figure 3—figure supplement 1C*, +FCCP). Taken together, the results shown in *Figures 1–3* reveal a single characteristic of the MFN2 R400Q mutation that distinguishes it from the CMT2A-linked MFN2 mutants: it suppresses Parkin recruitment to depolarized mitochondria.

## Mfn2 Q400 knock-in mice develop lethal perinatal cardiomyopathy

Accumulating evidence suggests that MFN-mediated mitochondrial fusion is essential to early embryonic heart development, and that MFN2-Parkin-mediated mitophagy is required for metabolic maturation of perinatal myocardium (*Kasahara et al., 2013*; *Gong et al., 2015*). Thus, MFN2 Q400-induced dysfunction of either fusion or mitophagy could underlie its apparent over-representation in human heart disease. The human MFN2 R400Q mutation is sufficiently rare that relating MFN2 Q400 dysfunction to a particular clinical manifestation would require linkage analysis in large families that were not identified in our FCM cohort. Alternately, we reasoned that manifestation of a cardiac phenotype unique to mice carrying the MFN2 R400Q mutation would provide unbiased evidence linking it to in vivo heart disease. To provide continuity with our in vitro studies comparing MFN2 R400Q to CMT2A-associated mutations at T105 and M376, we used CRISPR/Cas9 to engineer human mutations at all three MFN2 sites into the highly homologous mouse *Mfn2* gene (*Figure 4—figure supplement 1*).

Heterozygous knock-in mice from each line were viable, and therefore interbred to expand colonies. Mutant allele frequency of resulting offspring, expected to be ~50%, was significantly lower in Mfn2 T105M mice, and trended lower in Mfn2 R400Q mice (*Figure 4A*). Genotyping of 7-day-old pups revealed no homozygous mutant Mfn2 T105M mice and an abnormally low percentage of homozygous mutant Mfn2 R400Q mice, whereas mutant allele distribution of Mfn2 M376V mice followed Mendelian norms (*Figure 4B*). Analysis of embryos from timed crosses revealed no homozygous Mfn2 T105M mice as early as E12.5, recapitulating early embryonic lethality described in germline Mfn1 and Mfn2 knockout mice (*Chen et al., 2003*); heterozygous Mfn2 T105M mice had normal hearts and

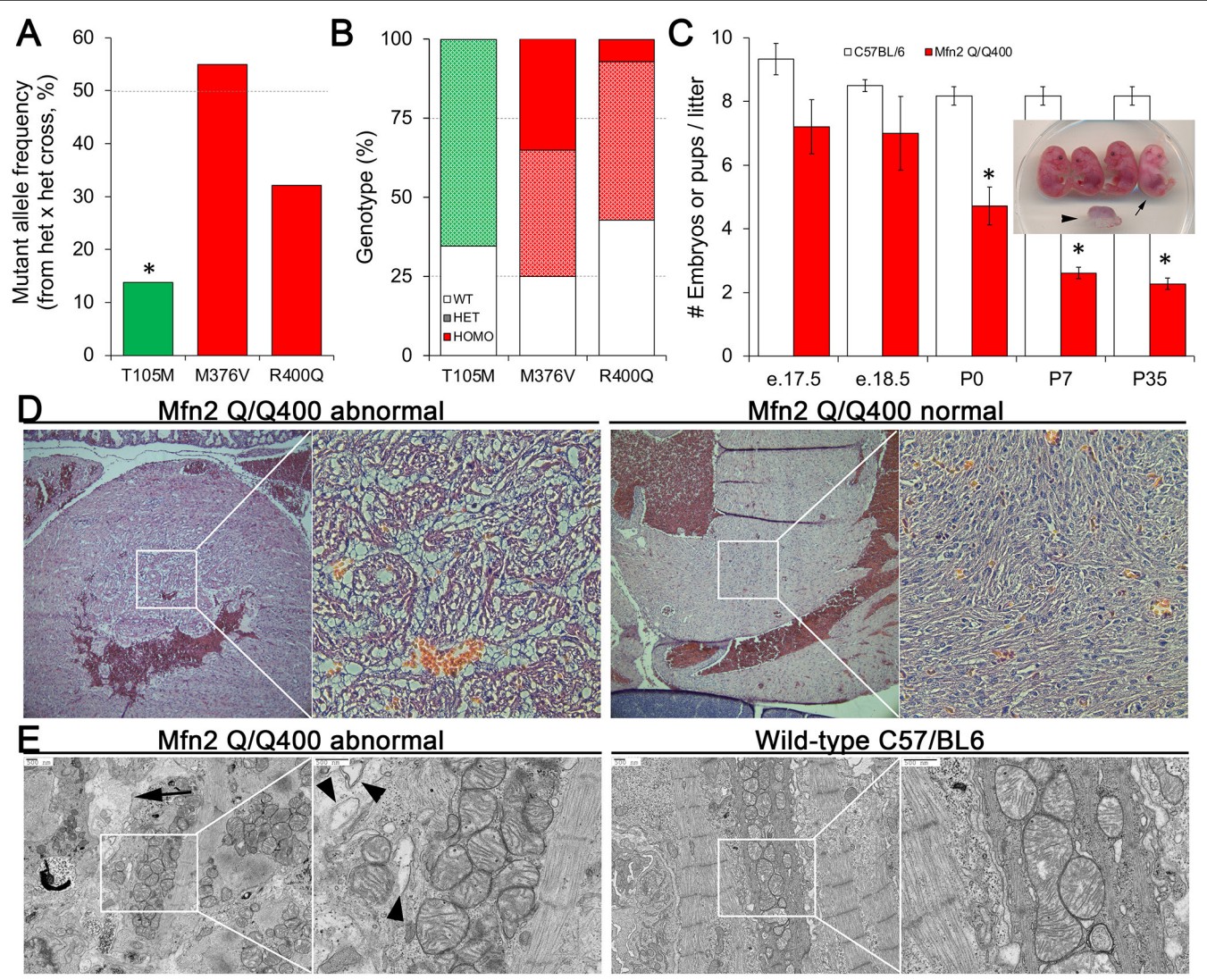

**Figure 4.** Perinatal cardiomyopathy in Mfn2 Q/Q400 mice. (**A, B**) Mutant allele frequencies (**A**) and genotypes (**B**) from heterozygous crosses of Mfn2 T105M, M376V, and R400Q knock-in mice (CRISPR/Cas9 knock-in strategies are given in the figure supplement). Expected values are shown as dotted lines. Homozygous mice were absent for T105M and fewer than expected for R400Q. (**C**) Fetal and early postnatal lethality of homozygous R400Q knock-in mice (Q/Q400) reported as mice/litter at different time points. Controls (white) are C57/Bl6. Inset shows an E18.5 Q/Q400 litter having a degenerated fetus (arrowhead) and a live, non-viable fetus (arrow). (**D**) Hematoxylin and eosin-stained sections of left ventricular myocardium from P0 Q/Q400 mice with (left) and without (right) cardiomyocyte vacuolization. (**E**) Ultrastructural studies of Q/Q400 myocardium revealing myofibrillar degeneration (arrow), empty 'ghost' mitochondria (arrowheads) and mitochondrial fragmentation; wild-type control is shown for comparison on the right. *p<0.05 vs. expected (**A**, Fisher's exact test) or control (**C**, Student's *t*-test).

The online version of this article includes the following figure supplement(s) for figure 4:

**Figure supplement 1.** CRISPR engineering of disease-linked human MFN2 mutants into the mouse genome.

**Figure supplement 2.** Quantitative mean data for transmission electron micrograph (TEM) studies of Mfn2 Q/Q400 mouse pups.

a neurological phenotype to be reported separately. Heterozygous and homozygous Mfn2 M376V mice likewise had normal hearts. We interrogated the apparent paucity of homozygous Mfn2 R400Q KI mice by interbreeding surviving homozygotes (Mfn2 Q/Q400 mice). Litter size from homozygous crosses, reflecting survival of Q/Q400 mice, was normal at E18.5. However, approximately one-third of Q/Q400 fetuses died between E18.5 and birth (P0), and another approximately one-third of Q/Q400 pups succumbed by P7, after which there was no further abnormal mortality (***Figure 4C***).

Histological examination of two litters of P0 Q/Q400 mice by a blinded Washington University veterinary pathologist revealed 'foci of cardiomyocyte vacuolation' in the myocardia of over half of the

hearts (*Figure 4D*). No other organs showed any histological or structural abnormalities. Ultrastructural examination revealed loss of myofibrillar integrity with mitochondrial scarcity, disorganization, fragmentation, and degeneration (*Figure 4E*, *Figure 4—figure supplement 2*) Thus, Mfn2 Q/Q400 mice develop a highly specific, incompletely penetrant late fetal/early perinatal cardiomyopathy.

## Transcriptional and metabolomic profiling of MFN2 Q400 cardiomyopathy identifies signatures specific for cardiac pathology

Transcriptional profiling can provide insights into the mechanisms underlying cardiac phenotypes produced by genetic variation (*Aronow et al., 2001*). We performed RNA sequencing on E18.5 MFN2 Q/Q400 mouse hearts and their WT controls; E18.5 was selected because there was little mouse drop-out at this time point (see *Figure 4C*). Unsupervised clustering of dysregulated Q/Q400 cardiac mRNAs (p-value<0.02, FDR < 0.05, fold change >2 vs. wild-type) segregated hearts into those having near-normal mRNA levels (designated nominal; Q400n) and those manifesting an abnormal transcriptional profile (designated anomalous; Q400a) (*Figure 5A*). Interrogation of mRNA levels of genes central to mitochondrial function, including respiratory enzymes encoded by the mitochondrial genome and nuclear-encoded mitochondrial biogenesis and mitochondrial dynamics genes (*Figure 5—figure supplements 1 and 2*), confirmed segregation between normal and abnormal hearts. Gene ontology (GO) analysis of dysregulated Q400a transcripts categorized mRNAs with lower expression in Q400a hearts overwhelmingly to metabolic and mitochondrial disease pathways (*Figure 5B*). By contrast, genes with increased expression in the same hearts were linked to multiple injury responses (*Figure 5B*), likely reflecting a compensatory reaction to myocardial degeneration. The distinction between Q400a and Q400n hearts was noteworthy for SOD2 and major cardiac expressed GPX mRNAs, respectively encoding mitochondrial superoxide dismutase and glutathione peroxidases that are central to mitigating mitotoxicity from reactive oxygen species (ROS) (*Figure 5C*).

The abnormal metabolic transcriptional signature of Q400a mouse hearts prompted metabolomics studies of individual newborn mouse hearts. As with gene expression, metabolomics segregated MFN2 Q/Q400 mice into Q400a and Q400n (*Figure 5D*). Metabolic differentiation between Q400a and Q400n was largely driven by abnormally low levels of a preferred metabolic substrate for fetal myocardium, lactate (*Lopaschuk et al., 1992*; *Piquereau and Ventura-Clapier, 2018*) in Q400a hearts. Levels of C2 fatty acid acylcarnitines and several amino acids were also diminished (*Figure 5—figure supplement 3*). Specific interrogation of metabolic pathway gene expression in MFN2 Q400 fetal hears revealed broad downregulation of mRNAs encoding fatty acid entry, glycolysis, tricarboxylic cycle, and oxidative phosphorylation pathway proteins (*Figure 5E–H*). Thus, the development of cardiomyopathy in Mfn2 Q/Q400 fetuses and neonates corresponds with signature abnormalities of myocardial metabolism.

## Cardiac myocyte mitophagy is suppressed by MFN2 Q400

Our in vitro studies revealed MFN2 Q400 to be defective in recruiting Parkin to mitochondria, suggesting that this mutation might adversely impact Parkin-mediated mitophagy. This notion gained support from our observation that the MFN2 Q400 fetal heart profile of abnormal fatty acid entry, glycolysis, tricarboxylic cycle, and oxidative phosphorylation pathway gene expression (see *Figure 5*) recapitulates transcriptional changes evoked by experimentally interrupting MFN2-Parkin-mediated mitophagy in perinatal hearts (*Gong et al., 2015*). Importantly, this mRNA profile has little similarity to the transcriptional signature provoked by suppressing mitochondrial fusion in embryonic myocardium (*Kasahara et al., 2013*). Given that MFN2 Q400 had only a modest effect on mitochondrial respiration (see *Figure 2C*), our findings suggested that the disturbances in myocardial metabolism induced by MFN2 Q400 might be a consequence of interrupting MFN2-Parkin-mediated mitophagy.

To better understand the impact of MFN2 R400Q on mitophagy in fetal heart cells, we expressed WT or mutant MFN2s in H9c2 rat cardiomyoblasts that retain features of embryonic cardiac myocytes (*Hescheler et al., 1991*; *Kuznetsov et al., 2015*; *Figure 6A*). First, we defined the time courses for FCCP-induced mitochondrial Parkin translocation (measured with mcParkin; *Narendra et al., 2008*), mitochondrial engulfment by autophagosomes (measured as mitochondria-LC3-GFP overlay; *Jackson et al., 2005*), and for mitochondrial delivery to lysosomes (measured using MitoQC; *Williams et al., 2017*) in H9c2 cells expressing WT MFN2. As expected, different stages of mitophagy progressed with different temporal patterns (*Figure 6B*). Remarkably, each of the stages of mitophagy was markedly

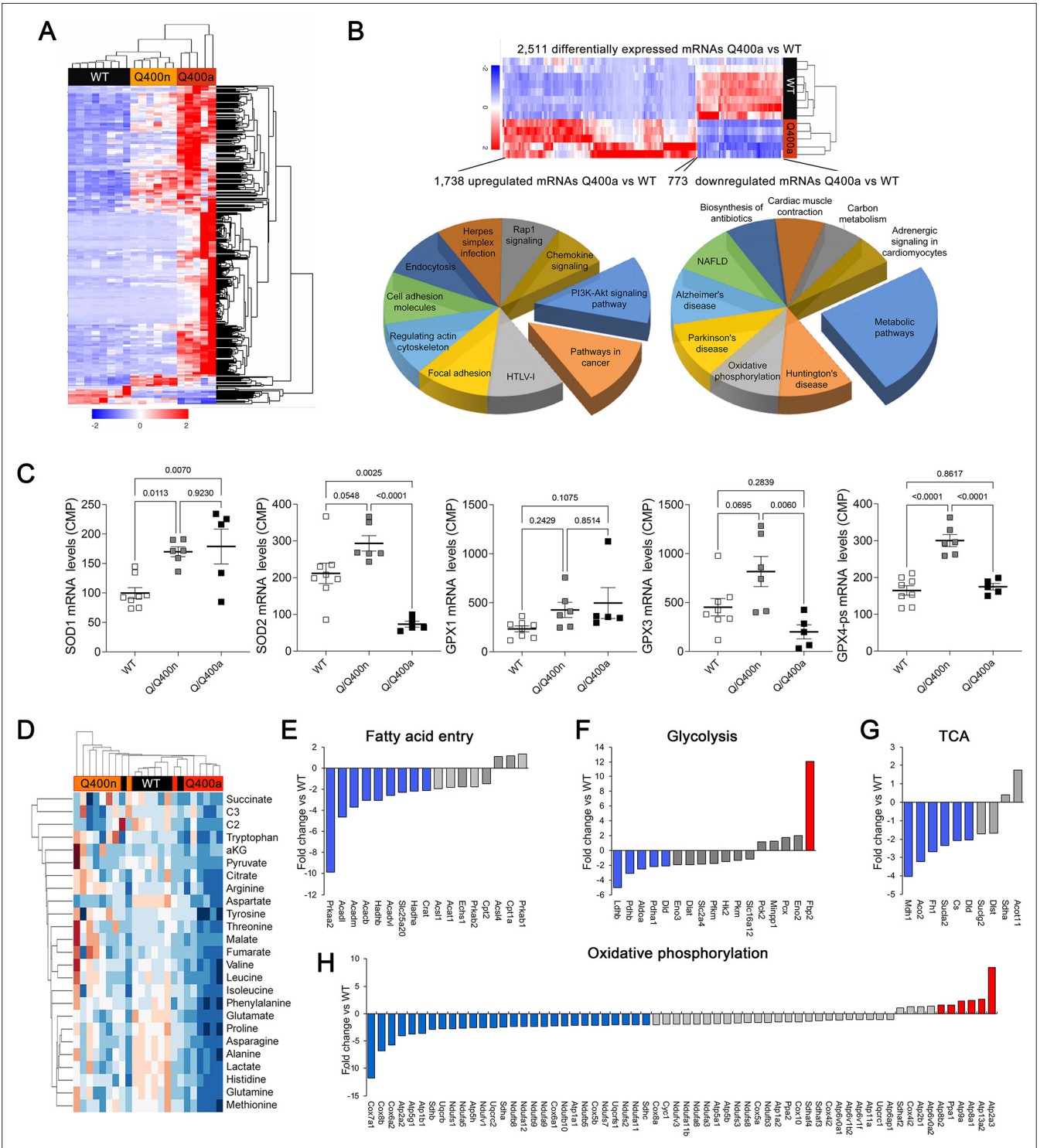

**Figure 5.** Transcriptional and metabolomic profiling of Mfn2 Q/Q400 mouse hearts identifies metabolic abnormalities characteristic of mitophagy defects. (**A**) Heat map of gene expression in individual Q400 and wild-type (WT) mouse hearts. (**B**) Heat map of anomalous Q400a mouse heart gene expression (top) and pie charts describing major KEGG functionally annotated pathway categories of up- (left) and downregulated (right) transcripts. (**C**) Individual heart mRNA levels for reactive oxygen species (ROS)-modulating enzymes. SOD, superoxide dismutase; GPX, glutathione peroxidase. Results shown are for the cardiac-expressed isoforms defined as WT CPM >100. Each point is an individual mouse heart. p-Values by ANOVA. (**D**) Metabolomics heat map showing unsupervised clustering of individual Q/Q400 and WT mouse hearts. (**E–H**) Relative expression of genes from indicated metabolic pathways. Blue is significantly decreased in MFN2 Q/Q 400a hearts; red is significantly increased; gray is no significant difference.

The online version of this article includes the following figure supplement(s) for figure 5:

*Figure 5 continued on next page*

Figure 5 continued

**Figure supplement 1.** Expression of mitochondrial DNA-encoded respiratory genes (top) and mitochondrial biogenesis genes (bottom) in late fetal Mfn2 Q/Q400 mouse hearts.

**Figure supplement 2.** Expression of mitochondrial dynamics factor mRNAs in late fetal Mfn2 Q/Q400 mouse hearts.

**Figure supplement 3.** Metabolite levels in Mfn2 Q/Q400 mouse pup hearts.

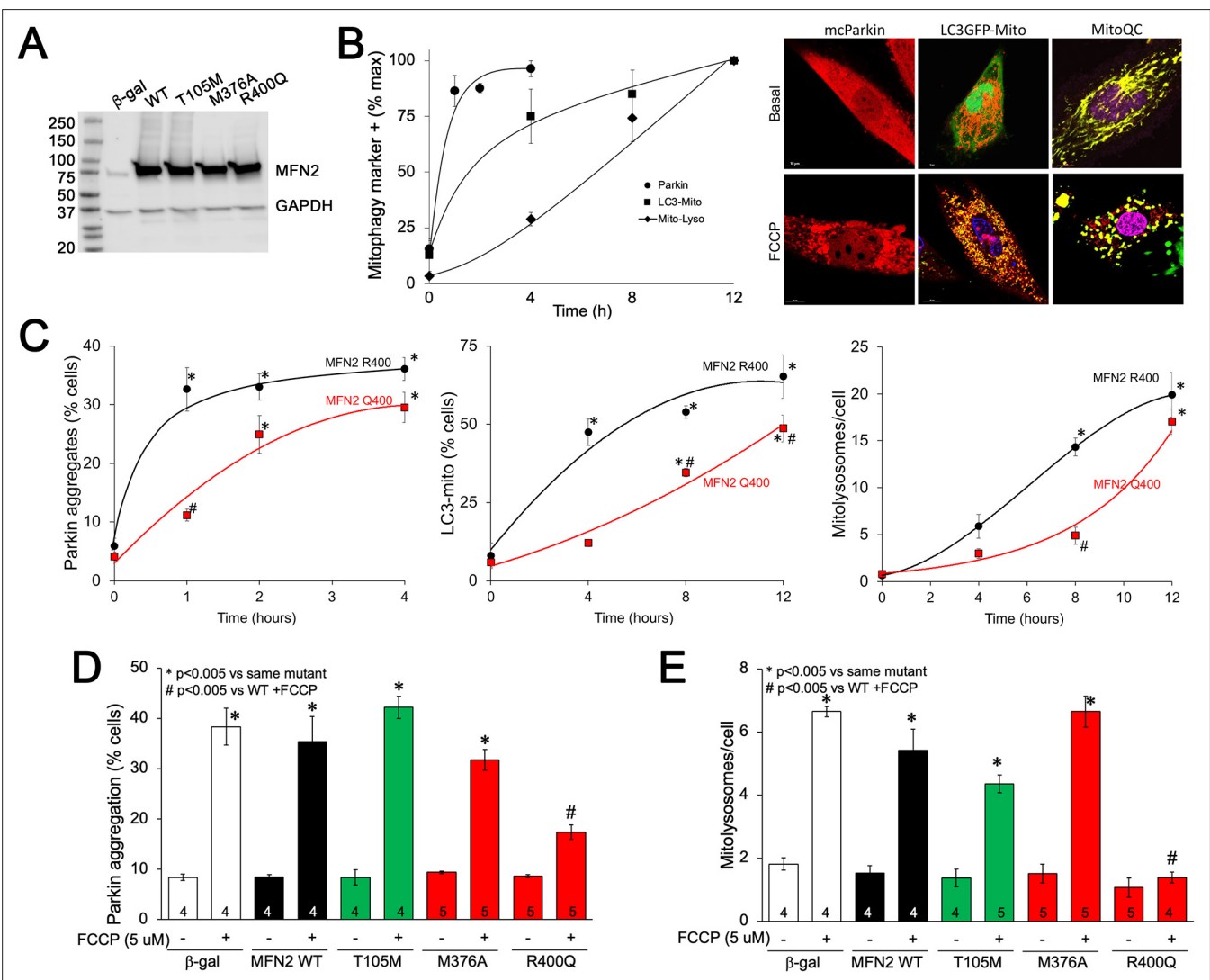

**Figure 6.** MFN2 Q400 uniquely suppresses early and late mitophagy in cardiomyoblasts. (**A**) Adenoviral expression of MFN2s in H9c2 cells. (**B**) Time courses of mitochondrial Parkin aggregation (circles), mitochondrial engulfment by autophagosomes (squares), and mitochondrial delivery to lysosomes (diamonds) in WT MFN2-expressing H9c2 cells after FCCP stimulation. Representative images are to the right. (**C**) Comparative time courses for mitochondrial Parkin aggregation (left), mitochondrial engulfment by autophagosomes (middle), and mitochondrial delivery to lysosomes (right) in MFN2 WT (black) and Q400 (red) expressing H9c2 cells. n = 3–4, *p<0.05 vs. pre-FCCP (time 0); #p<0.05 vs. WT at same time point (two-way ANOVA). (**D, E**) mcParkin aggregation 1 hr after FCCP treatment (**D**) and mitolysosome formation 8 hr after FCCP treatment (**E**) of H9c2 cells transduced with MFNs. Experimental n is shown in bars; stats by two-way ANOVA. Representative confocal images are shown in *Figure 6—figure supplement 1*.

The online version of this article includes the following source data and figure supplement(s) for figure 6:

**Source data 1.** Mitofusin 2 (75kDa) and GAPDH (36 kDa) expression level in H9c2 cells.

**Figure supplement 1.** Representative mcParkin (**A**) and MitoQC (**B**) confocal images showing mitophagy provoked by FCCP in H9c2 cardiomyoblasts expressing WT and disease-linked MFN2 mutants.

**Figure supplement 2.** Seahorse studies of oxygen consumption rate (OCR) in cultured H9c2 cardiomyoblasts expressing disease-linked MFN2 mutants.

delayed in H9C2 cells expressing MFN2 Q400 at an equivalent level (*Figure 6C*). Mitophagy slowly increased over longer time periods in MFN2 Q400 cells, likely reflecting MFN2-independent mitochondrial Parkin recruitment (*Ordureau et al., 2014*).

As in Mfn2 DKO cells (see *Figure 3C*), CMT2A-associated MFN2 T105M and M376A expressed in H9c2 cells promoted normal FCCP-stimulated Parkin recruitment to mitochondria, whereas Parkin aggregation was impaired in cardiomyoblasts expressing MFN2 R400Q (*Figure 6D*, *Figure 6— figure supplement 1A*); mitophagy measured as mitolysosome formation followed the same pattern (*Figure 6E*, *Figure 6—figure supplement 1B*). Mitophagy-defective MFN2 Q400 was less effective than WT MFN2 R400 in promoting H9c2 cardiomyoblast mitochondrial respiration (*Figure 6—figure supplement 2*), recapitulating its effects in MEFs (see *Figure 2C*). These findings further support the concept that the primary functional defect responsible for cardiac-specific abnormalities in MFN2 Q400 knock-in mice is impaired Parkin-mediated mitophagy.

## Mfn2 R400Q suppresses compensatory mitophagy induced by doxorubicin and increases sensitivity to doxorubicin-induced cardiomyopathy

Taken together, the above results point to defective mitophagy as a central mechanism underlying murine perinatal cardiomyopathy induced by the Mfn2 R400Q mutation. Based on prior reports (*Gong et al., 2015*), developmentally regulated Mfn2-Parkin-mediated mitophagy is essential for myocardial metabolic maturation in the perinatal period, whereas in adult hearts the same mitophagy mechanism mediates mitochondrial quality control (*Song et al., 2015*; *Song et al., 2017*). We asked whether, despite having normal cardiac function under basal conditions, surviving Q/Q400 mice might have a residual mitophagy defect manifested only in the face of an extrinsic mitochondrial insult. Doxorubicin, an anti-cancer chemotherapeutic with well-characterized dose-dependent cardiotoxic effects mediated in large part by mitochondrial dysfunction (*Wallace et al., 2020*; *Schirone et al., 2022*), is such an insult. Therefore, we evaluated dose-dependent doxorubicin-induced cardiomyoblast toxicity in cultured H9c2 cells. Expression of MFN2 Q400 increased overall doxorubicin-stimulated H9c2 cell death and TUNEL apoptosis marker labeling (*Figure 7A and B*), while suppressing mitochondrial Parkin recruitment and mitolysosome formation (*Figure 7C*). Compared to WT MFN2 R400, expression of MFN2 Q400 increased baseline mitochondrial ROS to levels comparable to those provoked by doxorubicin in normal cells (*Figure 7D*).

Administration of doxorubicin to mice provokes mitochondrial injury that can lead to dilated cardiomyopathy (*Dhingra et al., 2014*). We compared the cardiac response to doxorubicin between young adult wild-type (R/R 400), heterozygous (R/Q 400), and surviving homozygous (Q/Q 400n) Mfn2 R400Q knock-in mice. Left ventricular (LV) chamber dimension at end-diastole (LVEDD) and calculated LV mass increased, and LV ejection performance (fractional shortening; FS) decreased to a greater extent in homozygous Mfn2 Q/Q 400 knock-in mice than WT or heterozygous mice (*Figure 7E*). Likewise, gravimetric heart weights were greater in doxorubicin-treated Mfn2 Q/Q400 mice than the other groups (*Figure 7F*). As in cultured H9c2 cardiomyoblasts, the exaggerated Mfn2 Q/Q400 mouse cardiotoxic response to doxorubicin was associated with diminished reactive mitophagy (*Figure 7G*).

## Discussion

Our studies describe two major findings. First, we provide plausible evidence for a contribution of impaired MFN2-Parkin-mediated mitophagy to myocardial disease. Our conclusions are supported by a statistical genetic association in human cardiomyopathy, cardiac-specific phenotypes in knock-in mice expressing a candidate human mutation, and mechanistic studies defining the effects of the R400Q mutant MFN2 protein on mitophagy in cultured cardiomyoblasts and in vivo mouse hearts. Second, we describe a novel functional consequence of natural MFN2 mutation wherein an impaired ability to shift protein conformation leads to concomitant defects in both mitochondrial fusion and mitophagy, but a normal ability to promote mitochondrial transport. We surmise that the combination of mitophagy and fusion defects evokes cardiac disease, whereas combined mitochondrial motility and fusion defects described for many CMT2A-linked MFN2 mutations (including those studied herein) provoke peripheral neuropathy. This paradigm provides a mechanistic basis for tissue specificity of diseases evoked by MFN2 mutations.

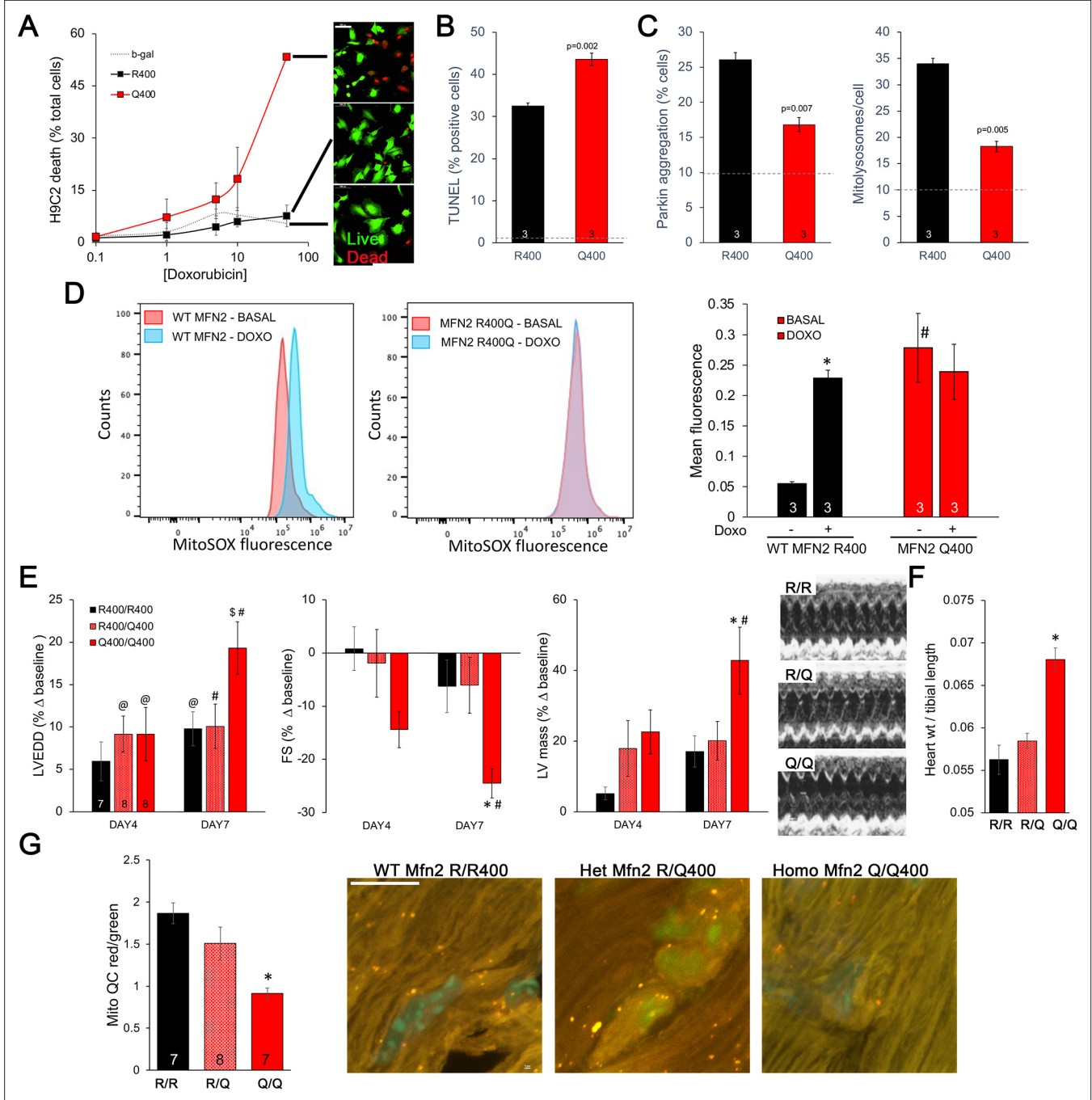

**Figure 7.** Mfn2 R400Q impairs reactive mitophagy and amplifies cardiomyocyte toxicity induced by doxorubicin in vitro and in vivo. (**A**) Cell death curves provoked by increasing concentrations of doxorubicin (overnight treatment) in WT MFN2 (R400) and MFN2 Q400 transduced H9*c*2 cardiomyoblasts. Representative live-dead images corresponding to each group are to the right; scale bar 100 um. (**B**) TUNEL labeling of apoptotic H9c2 cardiomyoblasts treated with 10 uM doxorubicin overnight. Dotted line is basal level in absence of doxorubicin. (**C**) Reactive mitophagy measured as Parkin aggregation (left) and mitolysosome formation (right) in doxorubicin-treated H9c2 cardiomyoblasts. Dotted lines show basal levels in absence of doxorubicin. (**D**) Flow cytometric analysis of doxorubicin effects on mitochondrial-derived ROS in H9c2 cardiomyoblasts expressing WT MFN2 R400 or mutant MFN2 Q400. Representative (of three independent experiments) histograms are on the left; mean group data are on the right. *p<0.05 vs. same group without doxorubicin; #p<0.05 vs. WT cells treated identically (ANOVA). (**E**) Echocardiography of WT (R/R400), heterozygous (R/Q400), and homozygous (Q/Q400) knock-in mice 4 and 7 d after receiving doxorubicin (20 ug/g, intraperitoneal injection). LVEDD is left ventricular (LV) end diastolic dimension, which increases with dilated cardiomyopathy; FS is LV fractional shortening, which decreases in dilated cardiomyopathy; calculated LV mass increases in dilated cardiomyopathy. Representative m-mode echos are to the right. (**F**) Gravimetric heart weights 7 d after doxorubicin, indexed to tibial length. (**G**) MitoQC visualization of LV cardiomyocyte mitolysosomes. Group mean quantitative data on the left; representative confocal images on the right. Mitolysosomes

*Figure 7 continued on next page*

Figure 7 continued

are bright orange spots; scale bar is 10 um. (**E–G**): *p<0.05 vs. WT at the same time; $p=0.05–0.075 vs. WT at the same time; #p<0.05 vs. same genotype pre-doxorubicin; @p=0.05–0.075 vs. same genotype pre-doxorubicin; statistical comparisons used one- (**F**, **G**) or two-way (**E**) ANOVA.

Conventional hypertrophic and dilated cardiomyopathies are caused by mutations of sarcomeric proteins (**Mazzarotto et al., 2020**). The current data suggest that myocardial-specific clinical disease can also be induced by mutational disruption of mitophagy. As an initiator/effector of both mitochondrial fusion and Parkin-mediated mitophagy, MFN2 orchestrates cardiomyocyte fate in embryonic (**Kasahara et al., 2013**) and perinatal hearts (**Gong et al., 2015**). Under normal circumstances, mitofusin–mitofusin binding mediates mitochondrial fusion, and mitofusin–Parkin binding promotes mitophagy; these are mutually exclusive functions regulated by PINK1 kinase (**Li et al., 2022**). Thus, non-phosphorylated WT MFN2 promotes MFN-MFN-mediated fusion, but cannot bind Parkin. MFN2 phosphorylation on S378 extinguishes fusion (**Rocha et al., 2018**) and on T111 and S442 enables Parkin binding to initiate mitophagy (**Chen and Dorn, 2013**). MFN2 phosphorylation on these residues and its control of mitochondrial fate are mechanistically linked by altered MFN2 protein partnering as a consequence of phosphorylation-induced changes in MFN2 conformation (**Dorn, 2020**). Here, we considered that the location of the MFN2 R400Q and M376A mutations near PINK1-phosphorylatable MFN2 S378 and S442 might alter MFN2 conformational switching. Indeed, FRET studies revealed impaired transitioning of both MFN2 R400Q and M376A from the open/fusion-active conformation to the closed/fusion-inactive but mitophagy-permissive conformation. By comparison, GTPase-defective MFN2 mutants readily changed conformation. Our data do not indicate that these two HR2 domain mutations are permanently fixed in either conformational state, but that their transitioning from one conformation to the other is retarded.

Lack of conformational malleability did not distinguish between the cardiomyopathy- vs. the neuropathy-associated MFN2 mutants. Neither did impaired catalytic activity. While most CMT2A-associated MFN2 mutations occur in the GTPase domain, MFN2 M376A/V is in the HR1 domain and, like R400Q, has normal GTPase activity. Here, mitochondria expressing GTPase-defective MFN2 mutants were both fragmented (from impaired fusion) and depolarized (from degeneration of the mitochondrial respiratory chain). Consistent with respiratory impairment, Seahorse studies revealed severely depressed mitochondrial oxygen consumption in MEFs and H9c2 cells expressing GTPase-defective MFN2 mutants. By contrast, MFN2 HR1 domain mutants M376A and R400Q had normal GTPase activity; mitochondria expressing these mutants, while fragmented, were normally polarized and with only mildly impaired respiration compared to WT MFN2. These findings suggest that mitochondrial depolarization (rather than just fragmentation) may play an important pathological role in CMT2A, but not cardiomyopathy.

Our data provide a scientific foundation supporting consideration of human genetic mutations that suppress cardiomyocyte mitophagy as candidate risk modifiers for myocardial disease. For MFN2 R400Q, the genetic association between this mitophagy-modifying mutation and human cardiomyopathy is intriguing. However, the mutation is sufficiently rare that it is not possible from genetic data alone to define its contribution to clinical heart disease. While absence of MFN2 Q400 homozygosity in humans could reflect severe early cardiac pathology as manifested by our KI mice, it may simply be the consequence of its rarity. It is worth considering that the low prevalence of this mutation compared to non-synonymous MFN2 mutations in the same HR1 domain, and its over-representation in individuals of African descent, could also reflect negative selection throughout modern human evolution of an ancient and harmful gene variant. Consistent with this idea, the split phenotype of Mfn2 Q400 KI mice, that is, Q400a mice that succumbed to cardiomyopathy during the critical perinatal developmental period vs. Q400n mice that survived to adulthood with apparently normal hearts, belies the common underlying defect in cardiomyocyte mitophagy. We surmise that Q400 mice suffering from developmental mitophagic cardiomyopathy as late fetuses or early pups suffered from metabolic abnormalities and failed to upregulate ROS-managing enzymes. By comparison, Q400 mice that did not develop the spontaneous perinatal cardiomyopathy were better compensated against baseline ROS and metabolic derangements, yet remained hyper-sensitive to the extrinsic mitochondrial stress induced by doxorubicin because reactive mitophagy was impaired.

Finally, the Mfn2 Q400 KI mouse phenotype may provide fresh insights into the relative importance of mitochondrial fusion vs. mitophagy in hearts. We CRISPR engineered three human MFN2

mutations into the mouse genome. All three MFN2 mutations, T105M, M376A/V, and R400Q, exhibited impaired fusogenicity, but only MFN2 T105M provoked mitochondrial depolarization, abnormally low respiration, and impaired mitochondrial motility. This mutation induced early embryonic lethality when homozygous and caused a CMT2A-like neuropathy (to be described elsewhere) that spared the heart in heterozygous mice. Both M376A/V and R400Q were impaired in their ability to change conformation, explaining loss of function despite normal catalytic GTPase activity. However, only R400Q adversely affected Parkin localization to depolarized mitochondria, and therefore mitophagy, and of the three knock-in mouse lines only R400Q mice exhibited cardiac involvement.

In conclusion, the current findings point to MFN2 and potentially other mitophagy signaling pathways as candidate genetic modifiers of cardiomyopathy. In the context of MFN2 mutations, these data extend the paradigm that specific functional defects exhibited by mutant MFN2 proteins determine organ system disease involvement.

# Materials and methods

**Key resources table**

| Reagent type (species) or resource | Designation | Source or reference | Identifiers | Additional information |
|---|---|---|---|---|
| Gene (*Mus musculus*) | *Mfn-2* | NCBI Gene | Gene ID: 170731 | MFN2 ENSMUSG00000029020 |
| Gene (human) | *MFN-2* | NCBI Gene | Gene ID: 9927 | MFN2 ENSG00000116688 |
| Genetic reagent (*M. musculus*) | Mfn2 R400Q knock-in mice | DornLab | Knock-in mice generated with point mutation in position 400 | |
| Genetic reagent (*M. musculus*) | Mfn2 T105M knock-in mice | DornLab | Knock-in mice generated with point mutation in position 105 | |
| Genetic reagent (*M. musculus*) | Mfn2 M376V knock-in mice | DornLab | Knock-in mice generated with point mutation in position 376 | |
| Genetic reagent (*M. musculus*) | C57BL/6J mice | C57Bl/6 | The Jackson Laboratory: 000664 | C57Bl/6 |
| Cell line H9c2 (rat myoblast) | Cardiomyoblast | ATCC | CRL-1446 | Rat embryonic cardiomyoblast |
| Cell line Mfn1/Mfn2 null (*M. musculus*) | Mfn1 and Mfn2 double knock out MEFs | ATCC | CRL-2993 | Murine embryonic fibroblasts |
| Cell line DRG neuronal cells | Adult mouse dorsal root ganglion | From 8- to 12-week-old C57BL/6J using enzymatic dissociation. | | *Franco et al., 2020* |
| Cell line mouse embryonic fibroblast | MEFs WT | From E.14.5 pups from C57BL/6J using enzymatic dissociation | https://pubmed.ncbi.nlm.nih.gov/18265203/ | |
| Transfected construct (human adenovirus type 5 dE1/E3) | Adenovirus Mito-Ds-Red2 | Signagen | Cat# 12259 | |
| Transfected construct (human adenovirus type 5 dE1/E3) | Adenovirus Cre-recombinase | Vector Biolabs | Cat# 1794 | |

*Continued on next page*

*Continued*

| Reagent type (species) or resource | Designation | Source or reference | Identifiers | Additional information |
|---|---|---|---|---|
| Transfected construct (human adenovirus type5 dE1/E3) | Ad-MFN2T105M | DornLab | | *Franco et al., 2022* |
| Transfected construct (human adenovirus type 5 dE1/E3) | Ad-MFN2R94Q | DornLab | | *Franco et al., 2022* |
| Transfected construct (human adenovirus type 5 dE1/E3) | Ad-MFN2R400Q | DornLab | Human adenovirus type 5 (dE1/E3), with Mfn2R400Q | |
| Transfected construct (human adenovirus type 5 dE1/E3) | Ad-MFN2K109A | DornLab | | *Rocha et al., 2018* |
| Transfected construct (human adenovirus type 5 dE1/E3) | Ad-MFN2M376A | DornLab | | *Rocha et al., 2018* |
| Transfected construct (human adenovirus type 5 dE1/E3) | Ad-MFN2WT | DornLab | | *Franco et al., 2022* |
| Transfected construct (human adenovirus type 5 dE1/E3) | Ad-hFRETMFN22WT | DornLab | | *Rocha et al., 2018* |
| Transfected construct (human adenovirus type 5 dE1/E3) | Ad-hFRETMFN2R94Q | DornLab | | *Franco et al., 2022* |
| Transfected construct (human adenovirus type 5 dE1/E3) | Ad-hFRETMFN2K109A | DornLab | Human adenovirus type 5 (dE1/E3), with Mfn2K109A | |
| Transfected construct (human adenovirus type 5 dE1/E3) | Ad-hFRETMFN2M376A | DornLab | | *Franco et al., 2022* |
| Transfected construct (human adenovirus type 5 dE1/E3) | Ad-hFRETMFN2T105M | DornLab | | *Franco et al., 2022* |
| Transfected construct (human adenovirus type 5 dE1/E3) | Ad-LC3-GFP | DornLab | Human adenovirus type 5 (dE1/E3), with LC3 for autophagy | |
| Transfected construct (human adenovirus type 5 dE1/E3) | Ad-mCherry-Parkin | Gift from Dr. Åsa Gustafsson | N/A | |

*Continued on next page*

*Continued*

| Reagent type (species) or resource | Designation | Source or reference | Identifiers | Additional information |
|---|---|---|---|---|
| Transfected construct (human adenovirus type 5 dE1/E3) | Ad-mitoQC | DornLab | Human adenovirus type 5 (dE1/E3), with mitoQC for mitophagy | |
| Antibody | Anti-Mfn-2 (mouse monoclonal) | Abcam | Cat# ab56889 | 1:1000 |
| Antibody | Anti-COX-IV (rabbit polyclonal) | Abcam | Cat# ab16056 | 1:1000 |
| Antibody | Anti-GAPDH (mouse monoclonal) | Abcam | Cat# ab82245 | 1:3000 |
| Antibody | Anti-β-actin (unconjugated monoclonal) | Proteintech | Cat# 66009-1 | 1:3000 |
| Antibody | Goat anti-rabbit IgG | Thermo Fisher | Cat# 31460 | 1:3000 |
| Antibody | Peroxidase-conjugated anti-mouse IgG | Cell Signaling | Cat# 7076S | 1:3000 |
| Sequence-based reagent | mMFN2R400Q knock-in mouse (forward) | DornLab | Knock-in mice generated with point mutation in position 400 | GGCATGTATGTGTAGGTCAGAG |
| Sequence-based reagent | mMFN2 R400Q knock-in mouse (reverse) | DornLab | Knock-in mice generated with point mutation in position 400 | CCCAGCTCCTCTGATTTGA |
| Sequence-based reagent | mMFN2 R400Q knock-in mouse (sequencing) | DornLab | Knock-in mice generated with point mutation in position 400 | CAGGTCTCCTTCCACCTTTAC |
| Sequence-based reagent | mMFN2 T105M knock-in mouse (forward) | DornLab | Knock-in mice generated with point mutation in position 105 | TGTTTACTTTGGAAGTAGGCAGTCT |
| Sequence-based reagent | mMFN2 T105M knock-in mouse (reverse) | DornLab | Knock-in mice generated with point mutation in position 105 | TTGTTCTTGGTGTCCCACTCTGA |
| Sequence-based reagent | mMFN2 T105M knock-in mouse (sequencing) | DornLab | Knock-in mice generated with point mutation in position 105 | TCGATGCTTAATGAGTGCTGCTGG |
| Sequence-based reagent | mMFN2 M376V knock-in mouse (forward) | DornLab | Knock-in mice generated with point mutation in position 376 | GTCCGGGCCAAGCAGATTGCAGAGGC CGTTCGTCTCATCATGGATTCCCTGCA CATCGCAGCTCAGGAGCAGCGGTGAGA |
| Sequence-based reagent | mMFN2 M376V knock-in mouse (reverse) | DornLab | Knock-in mice generated with point mutation in position 376 | GGTAGTAAAGAGCCTTTCTAGCTGAT |
| Sequence-based reagent | mMFN2 M376V knock-in mouse (sequencing) | DornLab | Knock-in mice generated with point mutation in position 376 | TTTCGAGAGGCAGTTTGAGGTAA |

*Continued on next page*

*Continued*

| Reagent type (species) or resource | Designation | Source or reference | Identifiers | Additional information |
|---|---|---|---|---|
| Commercial assay or kit | GTPase-Glo Assay | Promega | Cat# V7681 | |
| Chemical compound, drug | Mitofusin agonist TAT-MP-1 | Thermo Fisher | | Franco A et al., Nature 2016 |
| Chemical compound, drug | Mitofusin antagonist TAT-MP-1 | Thermo Fisher | | Franco A et al., Nature 2016 |
| Chemical compound, drug | L-Glutamine | Gibco | Cat# 25030-149 | |
| Chemical compound, drug | Goat serum | Jackson ImmunoResearch | Cat# 005-000121 | |
| Chemical compound, drug | Doxorubicin | Sigma | 44583 | |
| Chemical compound, drug | Glutaraldehyde | Electron Microscopy Science | Cat# 16216 | |
| Chemical compound, drug | MitoTracker Green | Thermo Fisher | Cat# M7514 | |
| Chemical compound, drug | FCCP | Thermo Fisher | Cat# C2920 | |
| Chemical compound, drug | Calcein AM | Thermo Fisher | Cat# C3100MP | |
| Chemical compound, drug | Hematoxylin-eosin | Sigma | Cat#: GHS116 & HT110116 | |
| Chemical compound, drug | TUNEL | Promega | Cat#: G3250 | |
| Chemical compound, drug | Hoechst | Thermo Fisher | Cat# H3570 | |
| Chemical compound, drug | MitoTracker Orange | Thermo Fisher | Cat# M7510 | |
| Chemical compound, drug | Tetramethylrhodamine ethyl ester | Thermo Fisher | Cat# T-669 | |
| Chemical compound, drug | MitoSOX Red | Thermo Fisher | Cat# M36008 | |
| Software, algorithm | ImageJ | A. Schneider | https://imagej.net/Sholl_Analysis | |
| Software, algorithm | Partek Flow | https://www.partek.com/partek-flow/ | N/A | |
| Software, algorithm | FlowJo 10 software | https://www.flowjo.com/solutions/flowjo/downloads/ | N/A | |

## Human study populations

Heart failure (1117 Caucasian; 628 African American) and non-failing control subjects (625 Caucasian; 628 African American) from the Cincinnati Heart Study (*Liggett et al., 2008*; *Matkovich et al., 2010*; *Cappola et al., 2011*) and hypertrophic cardiomyopathy (240 Caucasian; 46 African American) had all *MFN2* exons individually sequenced to identify DNA sequence variants. Additionally, 424 individuals (281 probands) with FCM from the DCM Precision Medicine Study (*Huggins et al., 2022*; *Trachtenberg et al., 2022*) were screened to detect any *MFN2* mutations suitable for linkage analysis; none were identified. Overall and race-specific *MFN2* allele frequencies were compared to non-failing controls from the Cincinnati Heart Study and to the gnomAD data set of 141,456 individuals from various racial backgrounds.

## Cell lines

*Mfn1/Mfn2* double null MEFs fibroblasts were purchased from American Type Culture Collection (ATCC Manassas, VA, CRL-2994). Normal MEFs were prepared by enzymatic dissociation from embryonic day E.13.5–14.5 C57BL/6J mice (The Jackson Laboratory, Cat# 000664). H9c2 rat cardiomyoblasts were purchased from American Type Culture Collection (ATCC, CRL-1446). MEFs and H9c2 cells were cultured at 37°C, 5% $CO_2$-95% air in Dulbecco's minimal essential medium (DMEM) containing glucose (4.5 g/l) (Thermo Fisher Scientific, 11965-084) with 10% (v/v) fetal bovine serum (FBS; Gibco, Gaithersburg, MD, Cat# 26140-079), 1× nonessential amino acids (Gibco, Cat# 11130051), 2 mM L-glutamine (Corning, NY, Cat# 34717007), and 1% (v/v) penicillin/streptomycin (Gibco, Cat# 15140-122). All cells were tested negative for mycoplasma. Adult mouse DRG neurons were prepared from 8- to 12-week-old C57BL/6J using enzymatic dissociation.

## Viral vectors

MFN2 mutants and FRET probes were generated using PCR mutagenesis, sequence verified, and sent to Vector BioLabs for custom adenovirus production. Ad-CMV-β-Gal was purchased from Vector BioLabs (Cat# 1080).

## Protein structure modeling

Hypothetical structures of human MFN2 were generated using I-TASSER and Chimera UCSF. The putative closed conformation is based on structural homology with bacterial dynamin-like protein (PDB: 2J69). The Mfn2 open conformation was downloaded from the AlphaFold protein structure database (AF-095140-F1). Domain coloring is as follows: green GTPase (AA 94-265); red HR1 (AA 338-421); and blue HR-2 (AA 681-757).

## Imaging

Live cell studies of MEFs, H9c2 cells, or DRGs to assess mitochondrial morphology, respiratory function, mitophagy, and motility, and transmission electron microscopy studies were performed as previously described (*Franco et al., 2016*; *Chen and Dorn, 2013*; *Misko et al., 2010* and *Rocha et al., 2018*). Briefly, confocal fluorescence microscopy was used to gauge mitochondrial aspect ratio (length/width) quantified based on MitoTracker Green staining using ImageJ; mitochondrial depolarization quantified as the number of green/total mitochondria in MitoTracker Green/TMRE double-stained images; mitochondrial Parkin aggregation quantified in cells expressing mcParkin with or without MitoTracker green; mitolysosome formation quantified as number of red dots per cell in Mito QC-stained images; mitochondrial engulfment by autophagosomes quantified as the ratio of yellow to total cells (green/red overlay) in cells expressing LC3-GFP and co-stained with MitoTracker Orange. Mitochondrial motility was assessed in processes of cultured mouse DRG neurons expressing mitochondrial targeted RFP using time-lapse confocal microscopy (1 frame/5 s).

*Whole mouse embryos or adult mouse hearts* were fixed with 4% formaldehyde solution in PBS, embedded in paraffin, and stained with hematoxylin-eosin (Sigma, GHS116 and HT110116) or TUNEL labeled using the manufacturer's protocol (Promega, G3250).

Transmission electron microscopy of mouse heart tissues fixed in 4% paraformaldehyde, 2.5% glutaraldehyde in 0.1 M sodium cacodylate buffer, pH 7.4, used thin sections stained with osmium tetroxide/uranyl acetate. A JEOL electron microscope (JEM-1400, JEOL, Tokyo, Japan) at 1500×–5000× direct magnification was used for image acquisition. Mitochondrial size and surface density were quantified using ImageJ (NIH).

## Western blotting

Seventy-two hours after adenoviral transduction (MOI 50) with MFN2 mutants, Mfn1/Mfn2 DKO MEFs and H9c2 were pelleted and lysed in cell extraction buffer on ice (Thermo Fisher Scientific, Cat# FNN0011) with 1 mM PMSF, protease inhibitor (Roche, Cat# 05892970001) and phosphatase inhibitor (Roche, Cat# 04906837001). Primary antibodies used were anti-Mfn2 (1:500, Abcam ab56889) and anti-β-actin (1:3000, Proteintech, Cat# 66009-1). Horseradish peroxidase (HRP)-conjugated anti-mouse IgG (1:3000, cs7076) was from Cell Signaling Technology.

## GTPase activity assay

*Mfn1/Mfn2* double-null MEFs were transduced with ad-MFN2 mutants (MOI 50). Mitochondria were isolated as described (*Dang et al., 2022*; *Dang et al., 2020*; *Detmer and Chan, 2007*; *Dhingra et al., 2014*), prepared on ice, and used fresh. Then, 100 μg of mitochondrial protein in triplicate was incubated in GTPase Buffer (Promega #V7681; Madison, WI) with 10 μM GTP and 1 mM DTT. Also, 1 μM MiM 111, 1 μM Chimera or DMSO (vehicle) were added as indicated and the reactants incubated at room temperature for 90 min in 96-well plates. The Promega GTPase-Glo assay kit was used to measure GTPase activity following the manufacturer's instructions. Luminescence was quantified on a Promega GloMax Luminometer.

## MFN2 FRET assays

Mfn1/Mfn2 double-null MEFs were transduced with FRET ad-MFNs. Then, 72 hr later mitochondria were isolated and FRET measured in a 96-well format as described (*Dang et al., 2020*).

## Mitochondrial respiration

Oxygen consumption rate (OCR) of mitofusin mutants expressed using adenoviral transduction of Mfn DKO MEFs or H9c2 cardiomyoblasts used a Seahorse XFe24 Extracellular Flux Analyzer (Seahorse Bioscience, Billerica, MA). Briefly, cells were plated on Seahorse XF24-well cell culture microplates for OCR measurements 72 hr after adenoviral transduction. Sensor cartridge hydration was performed overnight at 37°C. After basal respiration measurements, the following were autoinjected in sequence: 1 μM oligomycin to inhibit ATP synthase, 1 μM FCCP to uncouple oxidative phosphorylation, and 0.5 μM rotenone/antimycin A to abrogate electron transport (non-mitochondrial OCR). Each experiment averaged five or more replicate wells, and each experiment was repeated with a minimum of four biological replicates. ATP-linked respiration is basal OCR–OCR after oligomycin injection. Maximal mitochondrial respiration is OCR after FCCP–OCR after rotenone/antimycin A.

## Flow cytometry of mitochondrial ROS production

H9C2 cardiomyoblasts were transduced with adenoviral MFN2 constructs as described. Then, 72 hr later cells were co-stained with MitoSOX Red (5 μM for 10 min at 37°C) and MitoTracker Green FM 200 nM (Thermo Fisher, Cat# M36008 and #M7514, respectively). Fluorescence was analyzed on a ZE5 Flow Cytometer (Bio-Rad). Data were analyzed with FlowJo software version 10 and are presented as mean fluorescence intensity of five independent experiments.

## Creation of Mfn2 Q400 knock-in mice using Crisp/Cas9

The knock-in strategy is schematically depicted in *Figure 4—figure supplement 1A*. CRISPR gRNAs for in vitro testing were identified using http://crispr.mit.edu/. Mfn2 sgRNA was cloned into BbsI digested plasmid pX330 (Addgene # 42230). sgRNA activity was validated in vitro by transfection of NIH3T3 cells using ROCHE Xtremegene HP. Cell pools were harvested 48 hr later for genomic DNA prep, followed by Sanger sequencing of products spanning the gRNA/Cas9 cleavage site and TIDE analysis (https://tide.nki.nl/) of sequence trace files. T7 sgRNA template was PCR-amplified, gel-purified, and in vitro-transcribed with the MEGAshortscript T7 kit (Life Technologies). T7 Cas9 template was PCR-amplified, gel-purified, and in vitro-transcribed with the T7 mMessage mMachine Ultra kit (Life Technologies). After transcription, RNA was purified with Megaclear kit (Life Technologies). A 200 nt ssODN donor DNA with the mutation centered within the oligo was ordered from IDT as an ultramer oligo. Injection concentrations were 50 ng/μl Cas9, 25 ng/μl gRNA, and 20 ng/μl ssODN. Founders were identified using QIAGEN pyrosequencer and Pyromark Q96 2.5.7 software. All animal experiments were approved by institutional IACUC protocols. B6/CBA F1 mice at 3–4 wk of age (JAX Laboratories, Bar Harbor, ME) were superovulated by intraperitoneal injection of 5 IU pregnant mare serum gonadotropin, followed 48 hr later by intraperitoneal injection of 5 IU human chorionic gonadotropin (PMS from Sigma, HGC from Millipore USA). Mouse zygotes were obtained by mating B6/CBA stud males with superovulated B6/CBA females at a 1:1 ratio. Single-cell fertilized embryos were injected into the pronucleus and cytoplasm of each zygote using standard techniques.

- gRNA sequence: 5′ AAGAGCGGCAAGACCgGCTG 3
- Antisense ssODN sequence: 5′cttcattctcacCTGCCTTTCCACTTCCTCCGTAATCTGCTTAATTC GCAGCTTGTAGTCTTGAGCCAGGAGCTCCAGCTGCTTGTCAATAAACCTCAGCtGGTCTT

GCCGCTCTTCCCGCATTTCTAGGCAATAAACCctgagaggacaaaagcactgcttagaaacccgtgtcccca
caagagcaagcagagaagcca 3'

## Creation of Mfn2 T105M and M376V knock-in mice

CRISPR engineering was also performed at Cyagen, Inc to insert two CMT2A-associated mutations into the mouse genome. The overall procedures are similar to that described above.

*For Mfn2 T105M*: gRNA (forward strand) sequence was 5′ TTTCATCCTAGGACGAGCAATGG 3′. The p.T105M (ACG to ATG) in the donor oligo (see *Figure 4—figure supplement 1B*) was introduced into exon 6 by homology-directed repair. Two synonymous mutations, p.S106 (AGC to TCA) and p.S110 (AGC to TCG), were introduced simultaneously to prevent the binding and recutting of the sequence by gRNA after homology-directed repair; the SalI restriction site was used for genotyping.

For *Mfn2 M376V*: gRNA (forward strand) sequence was 5′ CTCCTGAGCTGCGATGTGCAGGG 3′. The p.M376V (ATG to GTG) in the donor oligo (see *Figure 4—figure supplement 1C*) was introduced into exon 12 by homology-directed repair. One synonymous mutation p.S378 (TCC to TCA) was also introduced to prevent the binding and recutting of the sequence by gRNA after homology-directed repair.

Pups were genotyped and correct integration of the fragment carrying the point mutation (p.T105M) or (p.M376V) was confirmed by Sanger sequencing. The correctly integrated mutant F0 founder mice were back-crossed with WT C57BL/6J mice to generate F1s. F1 heterozygous animals, also confirmed by Sanger sequencing, were interbred to build mouse colonies. All subsequent generation knock-mouse genotypes for all three lines were determined by Sanger sequencing.

## Omics studies

Total RNA from Mfn2Q/Q400 and C57BL/6J E.18.5 or P1 hearts was isolated using TRIzol Reagent (Cat# 15596-026, Ambion RNA Life Technologies) and treated with DNase I (Invitrogen, Cat# 18068015). mRNA library preparation and sequencing were performed at the Genome Technology Access Center (GTAC) of Washington University St. Louis, MO (USA) using the RiboErase protocol with indexing and pooled sequencing on a NovaSeq S4. Sequencing depth was 30M reads per sample. RNA-seq reads were aligned to the *Mus musculus* mm-10 assembly with STAR. Refseq was used as annotation model to quantify gene counts normalized to counts per million (CPM). Pair-wise comparisons between transcriptomic profiles from each experimental group were analyzed using GSA in Partek Flow and filtered a priori CPM ≥ 10 for at least one biological sample, FDR < 0.05, p-value<0.02, fold change >2, <-2. Unsupervised hierarchical clustering using Euclidean distance and average distance was performed using Partek Flow. KEGG Pathway analyses used Database for Annotation, Visualization and Integrated Discovery (DAVID v6.8). The GEO accession number is GSE214984.

Metabolomics analyses of acylcarnitines, organic acids, and amino acids in P0 hearts were performed as described (*Gong et al., 2015*).

## Doxorubicin experiments

In vitro – H9c2 cells were transduced with adenovirus encoding MFN2 R400 (WT) or Q400 at 100 MOI 48 hr prior to addition of doxorubicin; immunoblotting confirmed equal levels of MFN2 expression. MFN2-expressing cells were cultured in the presence of increasing doxorubicin concentrations (0.1–50 uM), or in the concentration specified, for 1, 12, or 24 hr as indicated. Live-dead staining used the kit from Invitrogen (Cat# L3224). TUNEL labeling used the DeadEnd Fluorometric kit from Promega (Cat# G3250). Mito QC mitolysosome labeling and mc Parkin aggregation studies are described above.

In vivo – 8–10-week-old male WT (R/R400), heterozygous (R/Q400 knock-in), and homozygous (Q/Q400 knock-in) mice underwent baseline echocardiographic assessments prior to receiving a single dose of doxorubicin (20 ug/g via intraperitoneal injection; *Dhingra et al., 2014*). Four days later, each mouse was administered adenovirus-Mito QC (3.3 × 10[11] particles in 100 ul). Echocardiography was performed 4 and 7 d post doxorubicin; mice were terminated after the 7-day echo study. Hearts were dissected, removed, placed in ice-cold physiological saline (0.9% NaCl), and weighed; heart weights were indexed to tibial length to account for individual variations in weight loss induced by doxorubicin. Mito QC staining was evaluated by confocal fluorescence imaging in 12 um frozen sections of OCT-embedded specimens.

## Data presentation and statistical analyses

Data are reported as means ± SEM. Sample number (n) indicates the number of independent biological samples. Two-group comparisons used Student's $t$-test; multiple group comparisons used one-way ANOVA with Tukey's post hoc test for individual statistical comparisons. Survival was evaluated by Kaplan–Meier analysis and the log-rank (Mantel–Cox) test. Comparisons of population MAF used chi-square testing. $p < 0.05$ was considered statistically significant.

## Acknowledgements

GWD conceived of and designed the research. GWD and AF wrote the manuscript. AF and JL performed mitochondrial function, GTPase and FRET assays, and knock-in mouse studies. RML and MS generated the CRISPR/Cas9 knock-in mouse. DPK and the Sanford-Burnham Medical Discovery Institute Metabolomics Core performed and analyzed the metabolomics studies. CdGS, MEM, ADS, AF JE, and XD analyzed RNAseq or human *MFN2* genotype data. RH, AJM, and GWD provided human *MFN2* sequencing from cardiomyopathy research cohorts. JE analyzed RNAseq in revised manuscript. All authors agree with their inclusion and place in the author list.

Supported by NIH R35 HL135736, R42 NS115184 and R42 NS113642 to GWD, R01 HL128349 to DPK, and by the Hope Center Transgenic Vectors Core at Washington University School of Medicine.

## Additional information

### Funding

| Funder | Grant reference number | Author |
| --- | --- | --- |
| National Institutes of Health | NIH R35 HL135736 | Gerald W Dorn |
| National Institutes of Health | R42 NS113642 | Gerald W Dorn |
| National Institutes of Health | R42 NS115184 | Gerald W Dorn |
| National Institutes of Health | R01 HL128349 | Daniel P Kelly |

The funders had no role in study design, data collection and interpretation, or the decision to submit the work for publication.

### Author contributions

Antonietta Franco, Resources, Data curation, Formal analysis, Validation, Investigation, Writing – original draft, Writing – review and editing; Jiajia Li, Resources, Formal analysis, Validation, Investigation; Daniel P Kelly, Resources, Data curation, Validation, Methodology; Ray E Hershberger, Resources, Data curation, Validation; Ali J Marian, Data curation, Validation, Methodology; Renate M Lewis, Moshi Song, Xiawei Dang, Mary E Mathyer, Validation, Methodology; Alina D Schmidt, Validation, Investigation; John R Edwards, Formal analysis; Cristina de Guzman Strong, Methodology; Gerald W Dorn, Conceptualization, Resources, Data curation, Formal analysis, Supervision, Funding acquisition, Investigation, Methodology, Writing – original draft, Writing – review and editing

### Author ORCIDs

Antonietta Franco http://orcid.org/0000-0002-5487-1800
Ray E Hershberger http://orcid.org/0000-0001-5683-6526
Xiawei Dang http://orcid.org/0000-0002-0343-7107
Gerald W Dorn https://orcid.org/0000-0002-8995-1624

### Ethics

All experimental procedures were approved by the Washington University in St. Louis, School of Medicine Animal Studies Committee; IACUC protocol number 22-0314 Exp:12/27/2025, .PI Gerald Dorn.

Decision letter and Author response
Decision letter https://doi.org/10.7554/eLife.84235.sa1
Author response https://doi.org/10.7554/eLife.84235.sa2

## Additional files

### Supplementary files
• MDAR checklist

### Data availability
Sequencing Data have been deposited in GEO under accession code GSE214984. - The putative closed conformation is based on structural homology with bacterial dynamin-like protein (PDB: 2J69). The Mfn2 open conformation was downloaded from the AlphaFold protein structure database (AF-095140-F1).

The following dataset was generated:

| Author(s) | Year | Dataset title | Dataset URL | Database and Identifier |
| --- | --- | --- | --- | --- |
| Dorn GW | 2023 | A human mitofusin 2 mutation can cause mitophagic cardiomyopathy | https://www.ncbi.nlm.nih.gov/geo/query/acc.cgi?acc=GSE214984 | NCBI Gene Expression Omnibus, GSE214984 |

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
