## [Editor Report]

The study presents a valuable finding on the link between mitofusin function (MFN2) and Parkin-mediated mitophagy, and that the combination of mitophagy and mitochondrial fusion defects is the basis of cardiomyopathy. The data based on the in vivo models are compelling and provide clinically relevant associations. This finding is important for understanding mitochondrial defects as a basis of heart pathologies.

---

## [Decision Letter]

**Decision letter after peer review:**

Thank you for submitting your article "A human mitofusin 2 mutation causes mitophagic cardiomyopathy" for consideration by *eLife*. Your article has been reviewed by 2 peer reviewers, and the evaluation has been overseen by a Reviewing Editor and Martin Pollak as the Senior Editor. The reviewers have opted to remain anonymous.

Essential revisions:

The reviewers find the paper valuable however they raised a number of points, which in our opinion are necessary to fully justify the conclusions. The most important part of revisions should provide additional evidence for a link between the mitofusin mutation R400Q and cardiomyopathy. Furthermore, additional support is needed to substantiate the defect in Parkin recruitment and mitophagy. The recommendations to the authors contain the requested or suggested experiments that would strengthen the findings and the importance of this nice study.

*Reviewer #1 (Recommendations for the authors):*

1. Replicate the association between MFN2 R400Q and cardiomyopathy using another dataset, e.g., the UK Biobank (which has a cohort design) or an independent dataset that has well matched cases and controls.

2. Report clearly the source and characteristics of the cases used in the association study described in Table 1.

3. In support of the claims in Figure 2A, assess Parkin recruitment to mitochondria in addition to the puncta formation. Assess downstream Parkin function such as a mitophagy assay would strengthen this assertion. If this can be done in a flow cytometry based assay (e.g., with mt-Keima) that requires less interpretation on the part of the experimenter that would increase confidence in the reproducibility of this line of experimentation.

4. Relatedly, the argument would be strengthened by in vivo data demonstrating decreased heart mitophagy in the KI mouse model (e.g., by injected of an AAV fluorescent mitophagy reporter such as mt-Keima, mito-GR, or mito-SRAI).

5. The authors should assess for cardiomyopathy in the heterozygous MFN2 R400Q mice or revise their claim that these data with knockin mouse support the idea that dominant mutations in MFN2 cause cardiomyopathy.

6. Page 9. Two CMT2A mutations are tested to establish a double disassociation between CMT2A-causing mutations and the R400Q variant which is proposed to be cardiomyopathy causing. However, the CMT2A mutations chosen all affect GTPase activity. What about CMT2A mutations that do not disrupt GTPase activity? Do they similarly not affect MFN2 conformational change in the FRET assay and Parkin recruitment.

7. In Figure 4H, couldn't these results mean that MFN2 R400Q is loss of function but not dominant negative whereas the other GTPase variants tested are dominant negative. What if these experiments are repeated in a cell line that lacks MFN2?

8. Line 22, page 3. I recommend adding a comma before "we."

9. Line 37, page 3. "[M]ean allele frequency" should be "minor allele frequency" as the authors previously states in line 23, page 3.

10. Line 34, page 6. I recommend deleting "Note-."

11. Figure 1F. Please describe how percentage of depolarized mitos were determined in the method section.

12. Figure 3C. Please describe what the numbers in bars indicate in the figure legend.

13. Figure 3D. Specify genotypes of the fetus.

14. Figure 5. I recommend mutating T111E and S442E on the adenoviral MFN2 R400Q construct to see if the mitophagy suppression can be rescued.

15. Missense mutations have been reported throughout different regions of MFN2 to cause CMT2A, including the same region as R400Q (eg. R400P and R397P). The argument that the unique combination of mitochondrial defects is the mechanistic basis for tissue specificity can be strengthened if a CMT mutation in the same region can be shown to lead to different mitochondrial defects and another mutation that affects confirmational malleability also leads to cardiac-specific disease.

*Reviewer #2 (Recommendations for the authors):*

1. The authors separated their Mfn2 Q400 homozygous knock-in mice into Mfn2 Q/Q400n (normal) and Mfn2 Q/Q400a (abnormal) and the data look interesting. However, they didn't clarify the criteria in distinguishing between two subtypes (by H&E staining results from Figure 3E maybe?). How about Mitofusin 2 expression level as well as the GTPase activity in these two subtypes? And based on their observation in Figure 3C, is Mfn2 Q/Q400n viable after P35 but not Mfn2 Q/Q400a? Moreover, what's the rationale of separating Mfn2 Q/Q400n and Mfn2 Q/Q400a in Figure 4- Supplemental 1 but combine them in Figure 4-Supplemental 2? In addition, for Figure 4D-G, are the data from Mfn2 Q/Q400a or Mfn2 Q/Q400n mice? This is probably the single most troubling aspect of the current study that needed to be rigorously dealt with.

2. Mechanistically, the authors showed that the Mfn2 Q400 mutant impaired mitophagy with decreased Parkin recruitment in vitro. However, the authors failed to address how this impairment in Parkin-mediated mitophagy results in perinatal cardiomyopathy in vivo. Indeed, this groups previous publication showed Parkin-mediated mitophagy directs perinatal cardiac metabolic maturation and inhibition of Parkin-mediated mitophagy caused lethal cardiomyopathy (doi: 10.1126/science.aad2459). However, how the current mutation in MFN2 relates is unclear, and direct evidence is needed to support the relevance of this manuscript and the mechanistic importance as to how the Q400 mutation is causing earl cardiomyopathy.

3. Mfn2 needs to be phosphorylated to serve as the Parkin receptor. While the authors showed no change in Mfn2 protein level in their mutants, have the authors examined Mfn2 phosphorylation level? This would seem to be critical.

---

## [Author Response]

Essential revisions:The reviewers find the paper valuable however they raised a number of points, which in our opinion are necessary to fully justify the conclusions. The most important part of revisions should provide additional evidence for a link between the mitofusin mutation R400Q and cardiomyopathy.

We thank the editors and reviewers for their detailed comments. We have performed a large amount of additional experimentation to strengthen the mechanistic link between MFN2 R400Q and cardiomyopathy: 1. We demonstrate that defective MFN2-mediated Parkin recruitment caused by the R400Q mutation evokes mitophagic dysfunction both in vitro in cultured H9c2 cardiomyoblasts and in vivo in Q/Q400 knock-in mice dosed with doxorubicin.

Furthermore additional support is needed to substantiate the defect in Parkin recruitment and mitophagy.

Although we wonder if the mito-QC data included in the original submission may have been overlooked, we now additionally report that: 1. MFN2 R400Q, but not CMT2A-associated MFN2 T105M or M376A/V, impairs FCCP-stimulated mitochondrial Parkin recruitment, autophagosomal engulfment of mitochondria, and mitochondrial delivery to lysosomes in H9c2 cardiomyoblasts; 2. MFN2 R400Q reduces mitochondrial Parkin aggregation and mitolysosome formation in H9c2 cardiomyoblasts treated with doxorubicin; and 3. MFN2 R400Q knock-in mice exhibit both a reduced myocardial mitophagy response to doxorubicin and increased susceptibility to doxorubicin-induced cardiomyopathy.

The recommendations to the authors contain the requested or suggested experiments that would strengthen the findings and the importance of this nice study.

As detailed in our responses to the reviewers, we’ve tried to address every concern. As you will appreciate, the manuscript has undergone major changes. While we have endeavored to retain all data from the original submission, to more effectively delineate and convey to readers the different properties of various disease-associated MFN2 mutations as specified by R1 in the 15^th^ Recommendation to Authors, the revised manuscript has been re-structured as a series of side-by-side comparisons between cardiomyopathy-associated MFN2 R400Q and two CMT2A-associated MFN2 mutations, MFN2 T105M in the GTPase domain and MFN2 M376A/V, in the HR1 domain. These specific comparisons are reported every scientific level: biophysical studies of protein conformation switching; enzymatic studies of catalytic GTPase activity; live-cell studies of mitochondrial fusion, polarization, motility and respiration in multiple cell types; mitophagy studies of mitochondrial Parkin localization and mitolysosome formation, and in vivo studies of mutant knock-in mice, including *data from two new (Mfn2 T105M and M376V) mouse KI models*. Previous data on artificial MFN2 mutants have been moved to the figure supplements. We believe that these changes have not only strengthened our conclusions, but increased the scientific and clinical impact of our findings.

Reviewer #1 (Recommendations for the authors):1. Replicate the association between MFN2 R400Q and cardiomyopathy using another dataset, e.g., the UK Biobank (which has a cohort design) or an independent dataset that has well matched cases and controls.

We have added genotyping data from the matched non-affected control population (n=861) of the Cincinnati Heart study to our analyses (page 4). The conclusions did not change.

2. Report clearly the source and characteristics of the cases used in the association study described in Table 1.

The details of our study cohorts were inadvertently omitted during manuscript preparation. As now reported on pages 3 and 4, the Cincinnati Heart Study is a case-control study consisting of 1,745 cardiomyopathy (1,117 Caucasian and 628 African American) subjects and 861 non-affected controls (625 Caucasian and 236 African American) (Liggett et al. *Nat Med* 2008; Matkovich et al. *JCI* 2010; Cappola et al. *PNAS* 2011). The Houston hypertrophic cardiomyopathy cohort (which has been screened by linkage analysis, candidate gene sequencing or clinical genetic testing) included 286 subjects (240 Caucasians and 46 African Americans) (Osio A et al. *Circ Res* 2007; Li L et al. *Circ Res* 2017).

3. In support of the claims in Figure 2A, assess Parkin recruitment to mitochondria in addition to the puncta formation. Assess downstream Parkin function such as a mitophagy assay would strengthen this assertion. If this can be done in a flow cytometry based assay (e.g., with mt-Keima) that requires less interpretation on the part of the experimenter that would increase confidence in the reproducibility of this line of experimentation.

The reviewer may have overlooked the studies reported in original Figure 5, in which Parkin localization to cultured cardiomyoblast mitochondria is linked both to mitochondrial autophagy (LC3-mitochondria overlay) and to formation of mito-lysosomes (MitoQC staining). These results have been retained and expanded to include MFN2 M376A in Figure 6 B-E and Figure 6 Figure Supplement 1 of the revised manuscript. Additionally, selective impairment of Parkin recruitment to mitochondria was shown in mitofusin null MEFs in current Figure 3C and Figure 3 Figure Supplement 1, panels B and C.

The in vitro and in vivo doxorubicin studies performed for the revision further strengthen the mechanistic link between cardiomyocyte toxicity, reduced parkin recruitment and impaired mitophagy in MFN2 R400Q expressing cardiac cells: MFN2 R400Q-amplified doxorubicin-induced H9c2 cell death is associated with reduced Parkin aggregation and mitolysosome formation in vitro, and the exaggerated doxorubicin-induced cardiomyopathic response in MFN2 Q/Q400 mice was associated with reduced cardiomyocyte mitophagy in vivo, measured with adenoviral Mito-QC (new Figure 7).

The LC3-mitochondrial overlay and MitoQC green-red shift data originally provided, now in Figure 6B-C and Figure 6 Figure Supplement 1. Parkin recruitment to/aggregation at mitochondria is now specifically demonstrated in Figure 3 Figure Supplement 1B, as well as extensively in the published literature.

4. Relatedly, the argument would be strengthened by in vivo data demonstrating decreased heart mitophagy in the KI mouse model (e.g., by injected of an AAV fluorescent mitophagy reporter such as mt-Keima, mito-GR, or mito-SRAI).

We are intrigued that in the previous comment the reviewer warns that murine phenocopies are not 100% predictive of human disease, and in the next sentence he/she requests that we show that the gene dose-phenotype response is the same in mice and humans. And, we again wish to note that we never argued that MFN2 R400Q “causes a dominant cardiomyopathy in humans.” Nevertheless, we understand the underlying concerns and in the revised manuscript we present data from new doxorubicin challenge experiments comparing cardiomyopathy development and myocardial mitophagy in WT, heterozygous, and surviving (Q/Q400n) homozygous Mfn2 R400Q KI mice (new Figure 7, panels E-G). Homozygous, but not heterozygous, R400Q mice exhibited an amplified cardiomyopathic response (greater LV dilatation, reduced LV ejection performance, exaggerated LV hypertrophy) and an impaired myocardial mitophagic response to doxorubicin. These in vivo data recapitulate new in vitro results in H9c2 rat cardiomyoblasts expressing MFN2 R400Q, which exhibited enhanced cytotoxicity (cell death and TUNEL labelling) to doxorubicin associated with reduced reactive mitophagy (Parkin aggregation and mitolysosome formation) (new Figure 7, panels A-D). Thus, under the limited conditions we have explored to date we do not observe cardiomyopathy development in heterozygous Mfn2 R400Q KI mice. However, we have expanded the association between R400Q, mitophagy and cardiomyopathy thereby providing the desired additional support for our argument that it can be a cardiomyopathy risk modifier.

5. The authors should assess for cardiomyopathy in the heterozygous MFN2 R400Q mice or revise their claim that these data with knockin mouse support the idea that dominant mutations in MFN2 cause cardiomyopathy.

We have performed the requested experiment, we do not see cardiomyopathy in heterozygous R400Q mice, and we have not revised the claims we actually made in the original manuscript:

6. Page 9. Two CMT2A mutations are tested to establish a double disassociation between CMT2A-causing mutations and the R400Q variant which is proposed to be cardiomyopathy causing. However, the CMT2A mutations chosen all affect GTPase activity. What about CMT2A mutations that do not disrupt GTPase activity? Do they similarly not affect MFN2 conformational change in the FRET assay and Parkin recruitment.

MFN2 M376A/V data have been added throughout to address this concern. The catalog of MFN2 mutations used clearly dissociates GTPase activity, conformational change, and Parkin recruitment. Please see data in Figures 1-3 and 6 and their accompanying supplements; these data are summarized in Author response table 1.

**Author response table 1. sa2table1:** 

MFN2 mutant	GTPase activity	Conformation change	Fusion	Polarization	Respiration	Mitophagy	Motility	KI phenotype
								
T105M	Inactive	Normal	Dominant suppressor	Dominant suppressor	Dominant suppressor	Normal	Dominant suppressor	CMT2A (het)Emb let (homo)
M376A/V	Normal	Impaired	Functional null	Normal	Functional null	Normal	Normal	Normal
R400Q	Normal	Impaired	Dominant suppressor	Normal	Functional null	Dominant suppressor	Normal	Cardiomyopathy (homo)

7. In Figure 4H, couldn't these results mean that MFN2 R400Q is loss of function but not dominant negative whereas the other GTPase variants tested are dominant negative. What if these experiments are repeated in a cell line that lacks MFN2?

Our results in the original submission, which are retained in Figures 1D and 1E and Figure 1 Figure Supplement 1 of the revision, exclude the possibility that R400Q is a functional null mutant for, but not a dominant suppressor of, mitochondrial fusion. We have added additional data for M376A in the revision, but the original results are retained in the main figure panels and a new supplemental figure.

Figure 1D reports results of mitochondrial elongation studies (the morphological surrogate for mitochondrial fusion) performed in Mfn1/Mfn2 double knock-out (DKO) MEFs. The baseline mitochondrial aspect ratio in DKO cells infected with control (b-gal containing) virus is ~2 (white bar), and increases to ~6 (i.e. ~normal) by forced expression of WT MFN2 (black bar). By contrast, aspect ratio in DKO MEFs expressing MFN2 mutants T105M (green bar), M376A and R400Q (red bars in main figure), R94Q and K109A (green bars in the supplemental figure) is only 3-4. For these results the reviewer’s and our interpretation agree: all of the MFN2 mutants studied are non-functional as mitochondrial fusion proteins.

Importantly, Figure 1E (left panel) reports the results of parallel mitochondrial elongation studies performed in WT MEFs, i.e. in the presence of normal endogenous Mfn1 and Mfn2. Here, baseline mitochondrial aspect ratio is already normal (~6, white bar), and increases modestly to ~8 when WT MFN2 is expressed (black bar). By comparison, aspect ratio is reduced below baseline by expression of four of the five MFN2 mutants, including MFN2 R400Q (main figure and accompanying supplemental figure; green and red bars). Only MFN2 M376A failed to suppress mitochondrial fusion promoted by endogenous Mfns 1 and 2. Thus, MFN2 R400Q dominantly suppresses mitochondrial fusion. We have stressed this point in the text on page 5, first complete paragraph.

8. Line 22, page 3. I recommend adding a comma before "we."

Done.

9. Line 37, page 3. "[M]ean allele frequency" should be "minor allele frequency" as the authors previously states in line 23, page 3.

Done.

10. Line 34, page 6. I recommend deleting "(Note-."

Done.

11. Figure 1F. Please describe how percentage of depolarized mitos were determined in the method section.

Done.

12. Figure 3C. Please describe what the numbers in bars indicate in the figure legend.

Done.

13. Figure 3D. Specify genotypes of the fetus.

Done.

14. Figure 5. I recommend mutating T111E and S442E on the adenoviral MFN2 R400Q construct to see if the mitophagy suppression can be rescued.

This is an interesting idea that we are exploring on a much larger scale by comparing the consequences of multiple MFN2 genetic mutations vs a variety of normal post-translational modifications in MFN regulation/dysregulation. Results describing functional interplay between genetic defect and post-translational modification, including many complementary experiments, will be published separately.

15. Missense mutations have been reported throughout different regions of MFN2 to cause CMT2A, including the same region as R400Q (eg. R400P and R397P). The argument that the unique combination of mitochondrial defects is the mechanistic basis for tissue specificity can be strengthened if a CMT mutation in the same region can be shown to lead to different mitochondrial defects and another mutation that affects confirmational malleability also leads to cardiac-specific disease.

This seemingly simple final request required us to perform the entire spectrum of comparative analyses between MFN2 R400Q vs multiple CMT2A-linked mutations in different MFN2 domains, including using MFN2 mutant KI mouse models that did not exist. The scope of this “recommendation” is breathtaking and, in the experience of the senior author, without precedent. Nevertheless, we have complied.

After careful consideration we selected CMT2A mutations different than those suggested by the reviewer because: 1. Proline substitutions provoke major structural changes in helical supercoils; 2. The suggested MFN2 mutants are vanishingly rare (perhaps single individuals or families); and 3. Neither of the suggested mutations are validated in CMT2A. Rather, we selected the prototypical MFN2 T105M GTPase domain and M376A/V HR1 mutations, each of which have been described multiple times in CMT2A patients. The revised manuscript has been restructured throughout to compare MFN2 R400Q to T105M and M376A/V.

The new results further support our original conclusion. To be clear however, the conclusion is not that a problem with conformational malleability confers cardiac disease as indicated by the reviewer, but that a problem with Parkin recruitment and downstream mitophagy evokes cardiomyopathy. Data in the revised manuscript, including analyses of new MFN2 T105M and M376V CRISPR/Cas9 KI mouse lines created for this purpose, strongly support this concept. Please see Author response table 1.

Reviewer #2 (Recommendations for the authors):1. The authors separated their Mfn2 Q400 homozygous knock-in mice into Mfn2 Q/Q400n (normal) and Mfn2 Q/Q400a (abnormal) and the data look interesting. However, they didn't clarify the criteria in distinguishing between two subtypes (by H&E staining results from Figure 3E maybe?). How about Mitofusin 2 expression level as well as the GTPase activity in these two subtypes? And based on their observation in Figure 3C, is Mfn2 Q/Q400n viable after P35 but not Mfn2 Q/Q400a? Moreover, what's the rationale of separating Mfn2 Q/Q400n and Mfn2 Q/Q400a in Figure 4- Supplemental 1 but combine them in Figure 4-Supplemental 2? In addition, for Figure 4D-G, are the data from Mfn2 Q/Q400a or Mfn2 Q/Q400n mice? This is probably the single most troubling aspect of the current study that needed to be rigorously dealt with.2. Mechanistically, the authors showed that the Mfn2 Q400 mutant impaired mitophagy with decreased Parkin recruitment in vitro. However, the authors failed to address how this impairment in Parkin-mediated mitophagy results in perinatal cardiomyopathy in vivo. Indeed, this groups previous publication showed Parkin-mediated mitophagy directs perinatal cardiac metabolic maturation and inhibition of Parkin-mediated mitophagy caused lethal cardiomyopathy (doi: 10.1126/science.aad2459). However, how the current mutation in MFN2 relates is unclear, and direct evidence is needed to support the relevance of this manuscript and the mechanistic importance as to how the Q400 mutation is causing earl cardiomyopathy.

If I understand, the reviewer is asking us to solidify the connection between the in vitro mitophagy defect of MFN2 Q400 and the in vivo cardiomyopathy in Mfn2 Q400 KI mice. This is similar to a major concern expressed by R1 who proposed that impaired mitochondrial fusion could be sufficient to explain the in vivo phenotype. We reasoned that the most rigorous way to do this was to extend our previous comparative in vitro analysis of a spectrum of MFN2 mutations to the in vivo mouse. The original submission included in vitro comparisons of MFN2 R400Q to the natural CMT2A MFN2 mutants R94Q and T105M (both GTPase domain) and to Mfn2 K109A that was laboratory engineered specifically to be GTPase defective. All of these data have been retained in revision, but R94Q and K109A results have been moved to figure supplements. We have added results for MFN2 M376A/V, a putative CMT2A mutant locus that like MFN2 R400Q is located within the HR1 domain. As noted in the table embedded in our responses to R1, T105M, M376A/V and R400Q represent natural MFN2 mutants that cover the entire spectrum of dysfunction.

By comparing the biochemical/biophysical, cellular and in vivo defects across all three mutants it was possible to exclude impaired fusion as the cause of observed cardiomyopathy (because T105M and M376A/V have similar fusogenic defects). Likewise, it is not possible that defective conformational shifting explains the cardiomyopathy (because M376A/V has the same impaired shape shifting as R400Q). Indeed, the only unique feature of R400Q is defective Parkin recruitment and mitophagy, and the only functional characteristic unique to R400Q that causes myocardial disease in KI mice relates to Parkin-mediated mitophagy. By the same reasoning, the only functional characteristic unique to T105M is mitochondrial dysmotility, which as we and others have described is mechanistically linked to neuropathy.

Regarding the comparison made by the reviewer with our previous work expressing mitophagy-defective MFN2 AA (T111, S442 engineered to non-phosphorylated Ala [A]) or MFN2 EE (same 2 amino acids engineered to Glu [E]): MFN2 AA has normal fusion but cannot bind Parkin (like R400Q) and when transgenically expressed from birth caused perinatal myopathy (like R400Q). (Side-by-side comparisons of MFN2 R400Q with MFN2 AA and EE in Figure 2 of the original submission have been moved to the Figure 3 Figure Supplement 1 to accommodate T105M and M376A/V data.) As noted in the manuscript (pg 9), the distinct metabolic and transcriptional signatures of the cardiomyopathy in Mfn2 AA mice are similar to those for the current R400Q mice. These similarities are consistent with the notion that interfering with cardiomyocyte mitophagy by interrupting MFN2-Parkin interactions cause mitophagic cardiomyopathy due to metabolic mismatch. We’d like to emphasize that the Gong et al. Science paper used cardiac-specific transgenic overexpression of an engineered Mfn2 from birth, whereas the current manuscript is a knock-in mouse wherein the (natural) Mfn2 R400Q mutant is expressed everywhere, but specifically impacted the heart.

Finally, we have now performed in vitro and in vivo studies using doxorubicin to stimulate compensatory cardiomyocyte mitophagy, which was suppressed by MFN2 Q400. The consequence in vitro and in vivo was increased sensitivity of MFN2 Q400 expressing cardiac cells to doxorubicin. These data are reported in a new manuscript section on pages 9 and 10, and new Figure 7. Together, we believe we’ve made a strong case that mitophagy underlies the cardiomyopathy provoked by MFN2 R400Q.

3. Mfn2 needs to be phosphorylated to serve as the Parkin receptor. While the authors showed no change in Mfn2 protein level in their mutants, have the authors examined Mfn2 phosphorylation level? This would seem to be critical.

This is a good question, which relates to the important issue of how post-translational mechanisms might be induced to compensate for genetic variation. Because Mfn2 is a poly-phosphorylated protein substrate for multiple kinases we don’t believe that interpretable data will be forthcoming from general determinations of Mfn2 phosphorylation level like Phos-tag SDS PAGE that we used a decade ago (Dorn and Chen, Science 2013). Rather, we are pursuing this idea by individual and combinatorial mutagenesis of functionally-relevant MFN2 phosphorylation sites (see Li et al. Front Cell Dev Biol 2022) in the context of our panel of disease-linked MFN2 mutations. This is an ongoing and extensive project whose results will be published in their entirety when completed. We can say that we see no evidence for altered MFN2 Q400 phosphorylation in otherwise normal cardiac cells.